# USP39 is essential for mammalian epithelial morphogenesis through upregulation of planar cell polarity components

Chiharu Kimura-Yoshida[1]✉, Kyoko Mochida[1], Shin-Ichiro Kanno[2] & Isao Matsuo [1,3]✉

Previously, we have shown that the translocation of Grainyhead-like 3 (GRHL3) transcription factor from the nucleus to the cytoplasm triggers the switch from canonical Wnt signaling for epidermal differentiation to non-canonical Wnt signaling for epithelial morphogenesis. However, the molecular mechanism that underlies the cytoplasmic localization of GRHL3 protein and that activates non-canonical Wnt signaling is not known. Here, we show that ubiquitin-specific protease 39 (USP39), a deubiquitinating enzyme, is involved in the sub-cellular localization of GRHL3 as a potential GRHL3-interacting protein and is necessary for epithelial morphogenesis to up-regulate expression of planar cell polarity (PCP) components. Notably, mouse *Usp39*-deficient embryos display early embryonic lethality due to a failure in primitive streak formation and apico-basal polarity in epiblast cells, resembling those of mutant embryos of the *Prickle1* gene, a crucial PCP component. Current findings provide unique insights into how differentiation and morphogenesis are coordinated to construct three-dimensional complex structures via USP39.

[1] Department of Molecular Embryology, Research Institute, Osaka Women's and Children's Hospital, Osaka Prefectural Hospital Organization, 840, Murodo-cho, Izumi, Osaka 594-1101, Japan. [2] IDAC Fellow Research Group for DNA Repair and Dynamic Proteome, Institute of Development, Aging and Cancer, Tohoku University, Sendai 980-8575, Japan. [3] Department of Pediatric and Neonatal-Perinatal Research, Graduate School of Medicine, Osaka University, Suita, Osaka 565-0871, Japan. ✉email: chiharu@wch.opho.jp; imatsuo@wch.opho.jp

Both growth and deformation of epithelial sheets drive morphogenesis such as gastrulation, tube formation, and body shape change. Coordination between movement, shape change, intercalation, division, and the death of epithelial cells drives these complex morphogenetic processes[1,2]. Notably, a change in cell shape is considered to be a consequence of a combination of the intrinsic properties of actomyosin and microtubule cytoskeletons, and the extrinsic properties of the biophysical environment[3]. However, the precise molecular mechanisms that direct a change in cell shape and subsequent epithelial morphogenesis are not fully understood.

Grainyhead-like (Grhl) factors can act as nuclear transcriptional factors and cytoplasmic regulators of planar cell polarity (PCP) components during epidermal differentiation and morphogenesis, respectively[4–9]. Consistent with previous reports, Grhl3 functions in the β-catenin–mediated specification of epidermal fate from uncommitted ectodermal progenitors in the nucleus (Fig. S1a, b)[8]. In addition, GRHL3 plays an important role in epithelial morphogenesis by regulating non-canonical Wnt signaling in the cytoplasm through a non-transcriptional mechanism (Fig. S1a, c)[9]. Notably, an interaction between cytoplasmic GRHL3 protein and PCP-related molecules is needed for acquisition of the biomechanical properties of matured epidermal cells, such as a change in cell shape and stiffness in the cell cortex (Fig. S1a–c)[9]. Thus, we proposed that cytoplasmic localization of GRHL3 from the nucleus upon epidermal differentiation is associated with a switch from canonical to non-canonical Wnt signaling (Fig. S1a–c)[9]. However, little is known about the regulatory mechanism involved whereby proteins can mediate the nuclear to cytoplasmic expression of GRHL3 and the cooperation of GRHL3 in the non-canonical Wnt pathway during mouse epidermal development.

The ubiquitin conjugation/deconjugation system is a critical regulatory process found in virtually all aspects of cellular phenomena, such as in the control of the cell cycle, cell fate, cell growth, antigen presentation, and cell signaling[10,11]. The removal of ubiquitin groups is controlled by a large family of deubiquitinating enzymes (DUBs), including ubiquitin-specific proteases (USPs). Deubiquitination is responsible for the timely removal of ubiquitin from substrates and, as such, can regulate protein function in a proteasome-dependent or -independent manner. Thus, DUBs are also critical for diverse cellular processes[12–18]. Although DUBs regulate cellular phenomena by controlling the assembly of multi-protein complexes, enzymatic activity, and subcellular localization, their mechanisms of action are not fully understood[19–22]. Of these, Usp39 encodes a conserved protein termed Sad1p in Saccharomyces cerevisiae and a 65-kD serine/arginine-rich (SR)-related protein in humans[23,24]. These two products are involved in the assembly of the spliceosome, the RNA splicing machinery[23,24]. Although ubiquitin-specific protease 39 (USP39) is suggested to lack deubiquitinase activity due to the absence of key active residues[23,24], it appears to deubiquitinate STAT1 protein and to upregulate the level of STAT1 protein in the cytoplasm[25]. However, the developmental roles of mouse Usp39 remain to be addressed.

In this study, we identified USP39 protein as a mediator of GRHL3 that activates PCP-related genes in the cytoplasm during epithelial morphogenesis. We found that USP39 is involved in the cytoplasmic localization of GRHL3 and is necessary for non-canonical Wnt signaling–mediated epithelial morphogenetic processes, such as changes in cell shape, apico-basal polarity, mesendoderm migration, axial elongation, and eyelid closure. Such USP39-dependent functions appear to be partly mediated by upregulating expression of PCP components through its deubiquitinating activity rather than the splicing process. These findings led us to propose that USP39 contributes to non-canonical Wnt signaling-dependent epithelial morphogenesis by upregulating expression of PCP components including cytoplasmic-localized GRHL3.

## Results

### Identification of USP39 as an interacting protein of GRHL3.

After initial epidermal differentiation, cytoplasmic GRHL3 that is relocalized from the nucleus triggers a change in cell shape and activates non-canonical Wnt signaling (Fig. S1a–c)[9]. Consequent activation of PCP molecules generates biomechanical forces, such as cell cortex stiffness for epithelial morphogenesis, by primarily enhancing actomyosin networks (Fig. S1a–c)[9]. To explore the molecular mechanisms that drive GRHL3 localization to the cytoplasm from the nucleus, we affinity-purified interaction partners of GRHL3 from MCF7 cells, a human cell line derived from breast adenocarcinoma, in which GRHL3 expression is observed in the nucleus and cytoplasm in THE HUMAN PROTEIN ATLAS (https://www.proteinatlas.org) (Fig. S1d–j). By expressing glutathione S-transferase (GST)-tagged GRHL3, or the GST-tag alone as a control, we identified protein complexes interacting with GRHL3 (Fig. S2a). Thereafter, several specific protein bands were further subjected to mass spectrometry. USP39 was subsequently determined to be a potential GRHL3-interacting protein, although according to the CRAPome: a contaminant repository for affinity purification–mass spectrometry data (http://www.crapome.org/), USP39 is a sticky protein found in ~15% of the protein pulldowns in that repository (Fig. S2a). To test the interaction between GRHL3 and USP39 within MCF7 cells under physiological conditions, we immunoprecipitated protein complexes in cell extracts using an antibody against GRHL3. We found that USP39 co-immunoprecipitated with GRHL3 (Fig. S2a).

To verify the interaction between USP39 and GRHL3 proteins, a visible immunoprecipitation (VIP) assay, a simple method to examine the interaction between two proteins without respective antibodies, was performed (Fig. S3)[26]. We constructed two expression vectors carrying enhanced green fluorescent protein (EGFP) fused to GRHL3, and RFP fused to USP39 in advance and, thereafter, co-transfected these into HEK293T cells (Fig. S3a). Consequently, we found that red fluorescent protein (RFP) fluorescence was undetectable in cell extracts from either EGFP-GRHL3 or RFP-USP39 transfectants but such fluorescence was obvious in extracts from both reporter constructs that were transfectants (Fig. S3b–e). Thus, the VIP assay demonstrated that nanobody beads that bind to EGFP–GRHL3 were able to interact with RFP–USP39 products (Fig. S3a, d, e). This finding suggests the possible interaction between GRHL3 and USP39 in vitro.

To confirm if GRHL3 localized in close proximity to USP39 in living cells, we analyzed protein–protein interaction using fluorescent protein fragments of a monomeric Kusabira-Green reporter system (mKG; CoralHueR Fluo-chase Kit) in MCF7 and NIH3T3 cells (Fig. 1a, b). Fragment pairs (mKG) without mGRHL3 or mUSP39 proteins failed to show a fluorescent signal (Fig. 1c–e, h–j). The fluorescent signals of reconstituted mKG fluorescence (green) were detected in both MCF7 and NIH3T3 cells when the mKGN-terminal fragment fused to mGRHL3, and mUSP39 fused to an mKGC-terminal fragment were co-expressed (Fig. 1f, g, k, l). These findings further support the idea that endogenous USP39 might bind GRHL3 in living cells.

To map which protein domains are responsible for the interaction between GRHL3 and USP39, we generated two series of constructs consisting of three domains of Grhl3 cDNA and two domains of Usp39 cDNA fused to GST, respectively, and examined binding activity to each other (Fig. S2b–e). We found that the CP2 and ubiquitin (Ub)-like domains of GRHL3 appeared to interact with the zinc finger (ZnF) domain of USP39 (Fig. S2d, e). These biochemical studies demonstrate that over-expressed GRHL3 can interact with USP39 via the CP2/Ub-like domains of GRHL3, and the ZnF domain of USP39.

Next, to determine the expression of USP39 protein in two additional cell lines (RT4, a human urinary bladder papilloma cell

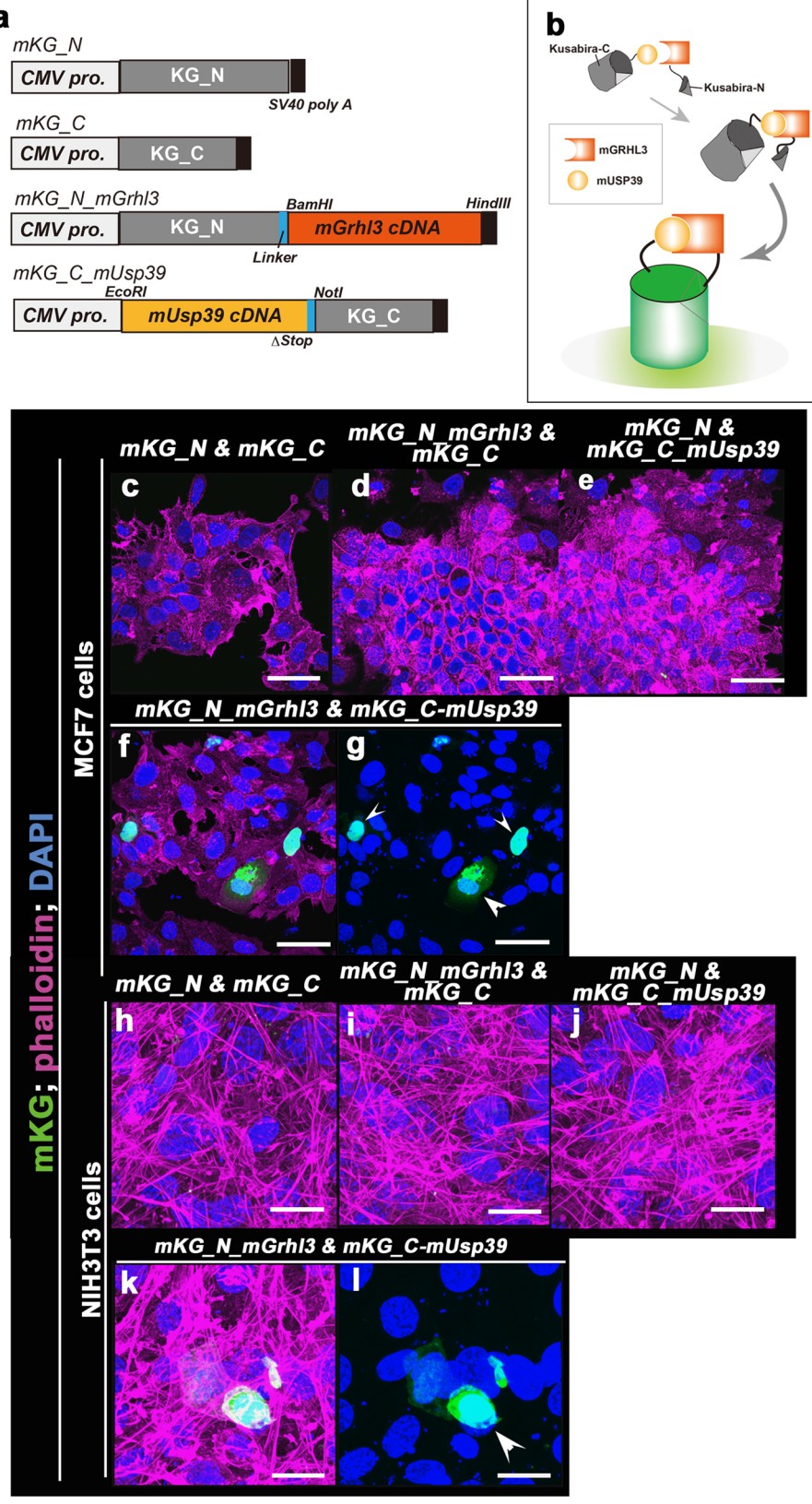

line, and MBT2 cells, a mouse cell line derived from a bladder cancer), we performed immunohistochemistry with anti-USP39 antibody (Fig. 2a–d). USP39 protein strongly localized in the cytoplasm and weakly in the nuclei of both RT4 and MBT2 cell lines (Fig. 2a–d). The specificity of immunofluorescent images with the anti-USP39 antibody was validated by Dicer-substrate small

interfering RNAs (DsiRNAs) of human USP39 (Dsi.hUSP39.13.1 and Dsi.hUSP39.13.2), a 27mer duplex RNA that possesses increased potency in RNA interference compared to traditional 21mer siRNAs (Fig. 2e–g). To elucidate the subcellular localization of USP39, we also examined expression of USP39 in MCF7 cells using live imaging after transfection of a HaloTag-hUSP39 plasmid

**Fig. 1 Detection of protein interaction between GRHL3 and USP39 using a monomeric Kusabira-Green system in vivo. a** Plasmids engineered for expression of N- or C-terminally monomeric Kusabira-Green reporter system (mKG) fragment-tagged protein fused to *mGrhl3* cDNA and *mUsp39*, respectively. **b** Schematic illustrations of the monomeric Kusabira-Green system which can detect protein–protein interactions as fluorescent signals using a protein fragment complementation method. **c–e**, **h–j** Negative controls of the mKG system in the absence of fused interacting modules in MCF7 (**c–e**) and NIH3T3 (**h–j**) cells. **f**, **g**, **k**, **l** A combination of complementary fusion proteins is capable of reconstituting mKG fluorescence when fused to interacting *mGrhl3* and *mUSP39* in the nucleus and cytoplasm of MCF7 (**f**, **g**; arrowheads) and NIH3T3 (**k**, **l**; arrowhead) cells. GRHL, grainyhead-like 3; USP39, ubiquitin-specific protease 39. Phalloidin (magenta) and DAPI (blue). Scale bars represent 20 μm (**h–l**) and 50 μm (**c–g**).

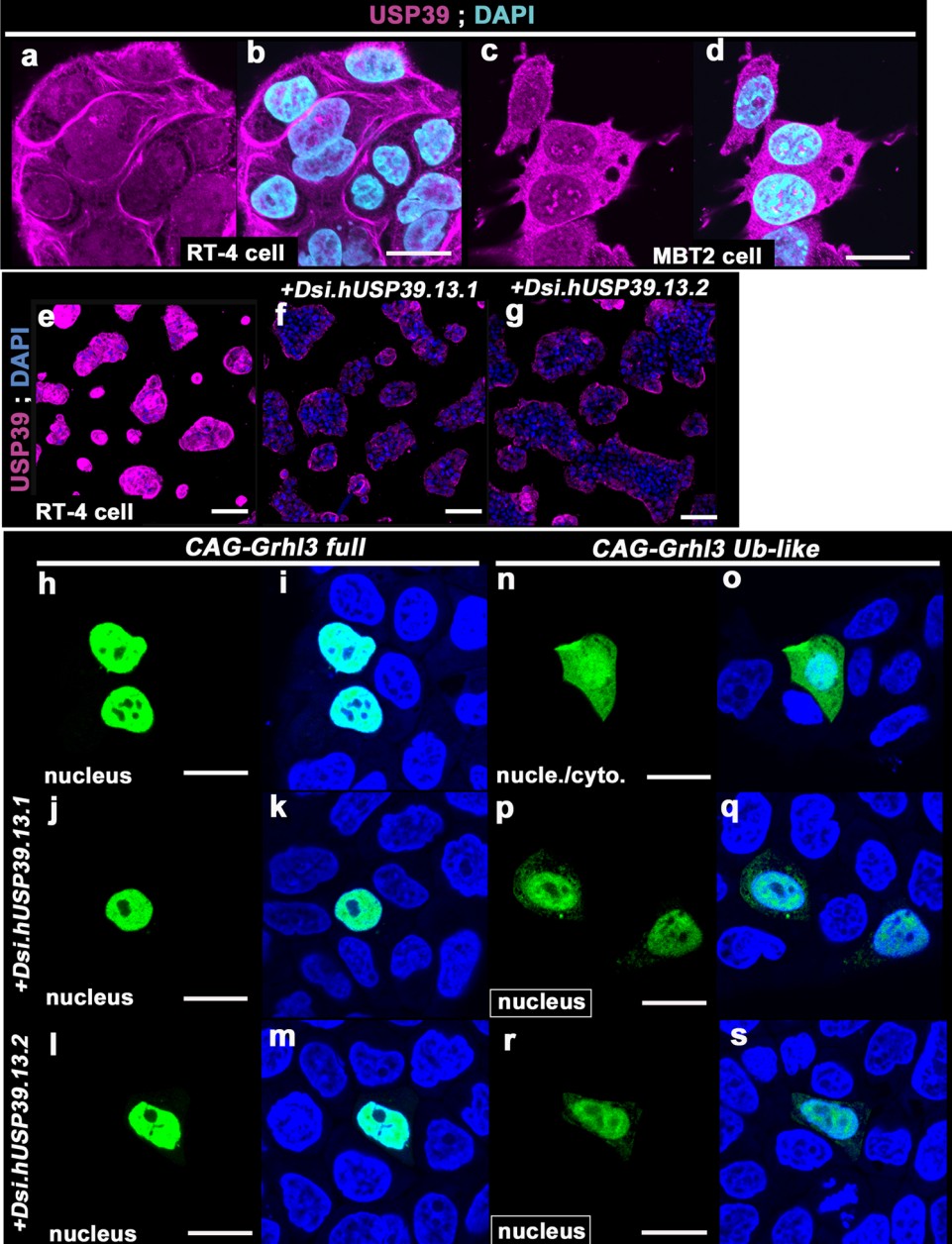

**Fig. 2 Localization of Ub-like domain of GRHL3 overlapping with USP39 expression. a–d** Ubiquitin-specific protease 39 (USP39) proteins are endogenously expressed strongly in the cytoplasm and weakly in the nucleus of RT4 (**a**, **b**) and MBT2 (**c**, **d**) cells. Ubiquitin-specific protease 39 (USP39) proteins (magenta) and 4,6′-diamidino-2-phenylindole (DAPI; light blue). **e–g** Immunohistochemistry of USP39 expression in RT4 cells exposed to *Dsi.hUSP39* for 24 h. USP39 proteins are present in most clumps of RT4 cells (**e**). USP39 protein is significantly decreased by treatment of cells with two different sequences of *Dsi.hUSP39* (**f**, **g**). **h–s** Subcellular localization of two types of grainyhead-like 3 (GRHL3) products: full length (**h–m**) and Ub-like domain (**n–s**) fused to enhanced green fluorescent protein (EGFP), respectively, and transfected into RT4 cells. The full-length GRHL3 fused to EGFP is localized to the nucleus (**h**, **i**), whereas a GRHL3–Ub-like domain fused to EGFP is localized in both the nucleus and cytoplasm (**n**, **o**). The Ub-like GRHL3 fused with EGFP localized mainly in the nucleus after transfection with *Dsi.hUSP39* (**p–s**), while full-length GRHL3 fused to EGFP was not affected by *Dsi.hUSP39* (**j–m**). Experiments were performed in duplicate. Scale bars represent 20 μm (**a–d**, **h–s**) and 100 μm (**e–g**).

and found that HaloTag fused USP39 protein localized mainly in the cytoplasm of MCF7 cells (Fig. S4a).

Since GRHL3 localizes in both the nucleus and cytoplasm depending on GRHL3 protein domains (Fig. S1d, g)[9], we tested whether endogenous USP39 expression can overlap with protein domains of GRHL3 fused to EGFP products in RT4 and MCF7 cells (Figs. 2h–s and S4b–f)[9]. The entire GRHL3-fused-to-EGFP (*GRHL3 full-EGFP*) protein localized mainly in the nucleus, which is concordant with endogenous GRHL3 localization as reported previously[9,27] (Figs. 2h, i and S4b; *Grhl3-full*). Endogenous USP39 localized in the cytoplasm of cells similar to HaloTag-fused USP39 expression (Fig. S4a–e: [red; Halo-tag, magenta; endogenous USP39]). Therefore, full-length GRHL3 appeared to be localized in the nucleus irrespective of endogenous USP39 expression (Fig. S4b; GRHL3-full; green, USP39; magenta). Similarly, the cytoplasmic-localized transactivation (TA) domain of GRHL3 appeared not to co-localize with endogenous USP39 (Fig. S4c). The CP2 domain of GRHL3, which localized in both the nucleus and cytoplasm, also appeared to not co-localize with USP39; however, several fractions of cytoplasmic GRHL3 overlapped with USP39 expression, supporting the aforementioned data on biochemical interaction via the CP2 domain of GRHL3 (Fig. S4d). Moreover, the Ub-like domain of GRHL3 largely overlapped with endogenous USP39 in the cytoplasm of RT4 and MCF7 cells (Figs. 2n, o and S4e; Grhl3–Ub-like fused with EGFP [green] and USP39 [magenta]). These aforementioned findings, together with biochemical results, suggest that over-expressed USP39 interacts with GRHL3 in the cytoplasm mainly through the Ub-like domain of GRHL3.

Given that overlapped expression between USP39 and GRHL3 appears to be evident in the cytoplasm (Fig. S4), it can be hypothesized that USP39 may be responsible for the cytoplasmic localization of GRHL3. To evaluate if reduced USP39 expression could affect GRHL3 localization, we exploited *DsiUSP39*, and the siRNA of *USP39*, and analyzed EGFP-fused GRHL3 expression in RT4 and MCF7 cells, respectively (Figs. 2j–m, p–s and S4f). Consequently, although *Grhl3-full-EGFP* mainly localized in the nucleus irrespective of USP39 knockdown in RT4 cells (Fig. 2j–m), localization of the GRHL3-Ub–like domain was reduced in the cytoplasm of USP39 knockdown cells (Figs. 2p–s and S4f). Taken together, these findings indicate that USP39 is necessary for the cytoplasmic localization of the Ub-like domain of GRHL3 in RT4 and MCF7 cells. This suggests that USP39 is involved in the localization of GRHL3 in the cytoplasm.

**USP39 contributes to non-canonical Wnt signaling during epidermal differentiation**. Considering that USP39 mediates the cytoplasmic localization of GRHL3, it can be assumed that USP39 expression would be observed during epidermal maturation. We had previously exploited the in vitro culture of embryonic stem (ES) cells whereby embryoid bodies (EBs) differentiated into large and mature (LM-) epidermal cells after the overexpression of *Grhl3* cDNA (Fig. S1a–c)[9]. Notably, GRHL3 expression induced large, multinucleated, and F-actin–enriched LM-epidermal cells (Figs. 3a and S1c). GRHL3 in epidermal cells initially localized in the nucleus and translocated to the cytoplasm during epidermal maturation (Fig. S1a)[9]. Such cytoplasmic localization of GRHL3 is crucial to induce LM-epidermal cells[9]. As we expected, the localization of USP39 protein was observed in the cytoplasm of LM-epidermal cells, in particular, F-actin–enriched domains (Fig. 3a; USP39 [green], F-actin [red]). Thus, consistent with the cytoplasmic localization of USP39 in RT4, MBT2, and MCF7 cell

lines (Figs. 2a–d and S4a), USP39 appears to be localized in the F-actin–enriched cytoplasm of LM-epidermal cells (Fig. 3a).

To validate the roles of USP39 in epidermal maturation from ES cells through EBs, we knocked down USP39 expression in EB cells and analyzed the formation of LM-epidermal cells driven by *Grhl3* overexpression (Fig. 3b, c). We found that USP39 knockdown in EB cells by *DsiRNA of Usp39* repressed the formation of LM-epidermal cells as compared to those from control EB cells by *Grhl3* overexpression (the number of LM-epidermal cells induced EB colonies/the number of all EB colonies, *CAG-Grhl3*; $n = 30/30$, *CAG-Grhl3* + *DsiRNA mUsp39*; $n = 14/20$, $p = 0.014 < 0.05$; Fig. 3b, c). These findings indicate that USP39 is involved in the formation of LM-epidermal cells from EBs driven by GRHL3 overexpression.

GRHL3 exerts two distinct functions that are closely linked to the specification of epidermal fate in the nucleus and epithelial morphogenesis in the cytoplasm: the former function is mediated by nuclear-localized GRHL3 as a downstream transcription factor of β-catenin and the latter is mediated by cytoplasmic-localized GRHL3 involving activation of non-canonical Wnt signaling (Fig. S1a–c)[8,9]. From the above findings, it can be hypothesized that USP39 might be involved in epithelial morphogenesis by upregulating PCP components in cooperation with GRHL3. To test this, two cDNA vectors, the constitutive active form of *β-catenin* (*β-catenin S37A*) (Fig. 3d, e) and USP39 cDNA (*USP39 full-length cDNA fused to RFP*; Fig. 3f) were transfected simultaneously into ES cells (Fig. 3g, h). The *β-cateninS37A* vector alone failed to induce LM-epidermal cells, although *β-cateninS37A* induced cytokeratin 8 (TROMAI)-positive small and immature epidermal cells indicating that epidermal differentiation can occur (Figs. 3d, e and S1b). Additionally, *CAG-RFP-Usp39* was not able to induce either LM-epidermal cells or small immature epidermal cells efficiently (Fig. 3f). However, the simultaneous transfection of both plasmids (*β-cateninS37A* and *Usp39* cDNAs) was able to induce the formation of LM-epidermal cells without *Grhl3* cDNA (Fig. 3g, h). Additional expression analyses with epidermal and PCP-related markers revealed that LM-epidermal cells induced by *β-cateninS37A* and *Usp39* cDNAs expressed mature epidermal markers, such as TFAP2A, TFAP2B, and keratin 17/19 (Fig. 3i–k), and showed elevated PCP-related activity as marked by expression of SCRIB and phosphorylated myosin light chain (Fig. 3l, m), similar to that induced by overexpressing *Grhl3* cDNA[9]. Similarly, USP39 alone was able to induce LM-epidermal cells in the presence of Wnt agonist, a chemical canonical Wnt activator, in culture (Fig. 3o, p). Since both canonical and non-canonical Wnt pathway signaling is necessary for the formation of LM-epidermal cells[9], and USP39 was able to induce LM-epidermal cells when canonical Wnt signaling alone was activated (Fig. 3g–p), these findings indicate that USP39 can direct epithelial morphogenesis by enhancing the enrichment of actomyosin networks via activation of non-canonical Wnt signaling.

**Usp39-deficient mouse embryos fail to form the primitive streak or express PCP components correctly**. Since roles played by USP39 in mouse morphogenesis have not yet been studied, we generated *Usp39*-deficient mice and analyzed their phenotypes (Figs. 4, 5, and S5). We produced *Usp39* knockout mice by targeting exon 1 using a CRISPR/Cas9 gene editing method and successfully obtained three independent knockout mouse lines (F0-12-1, F0-18-1, and F0-8; Fig. 4a). We found that the phenotypes of these mutant lines appeared to be identical and chose

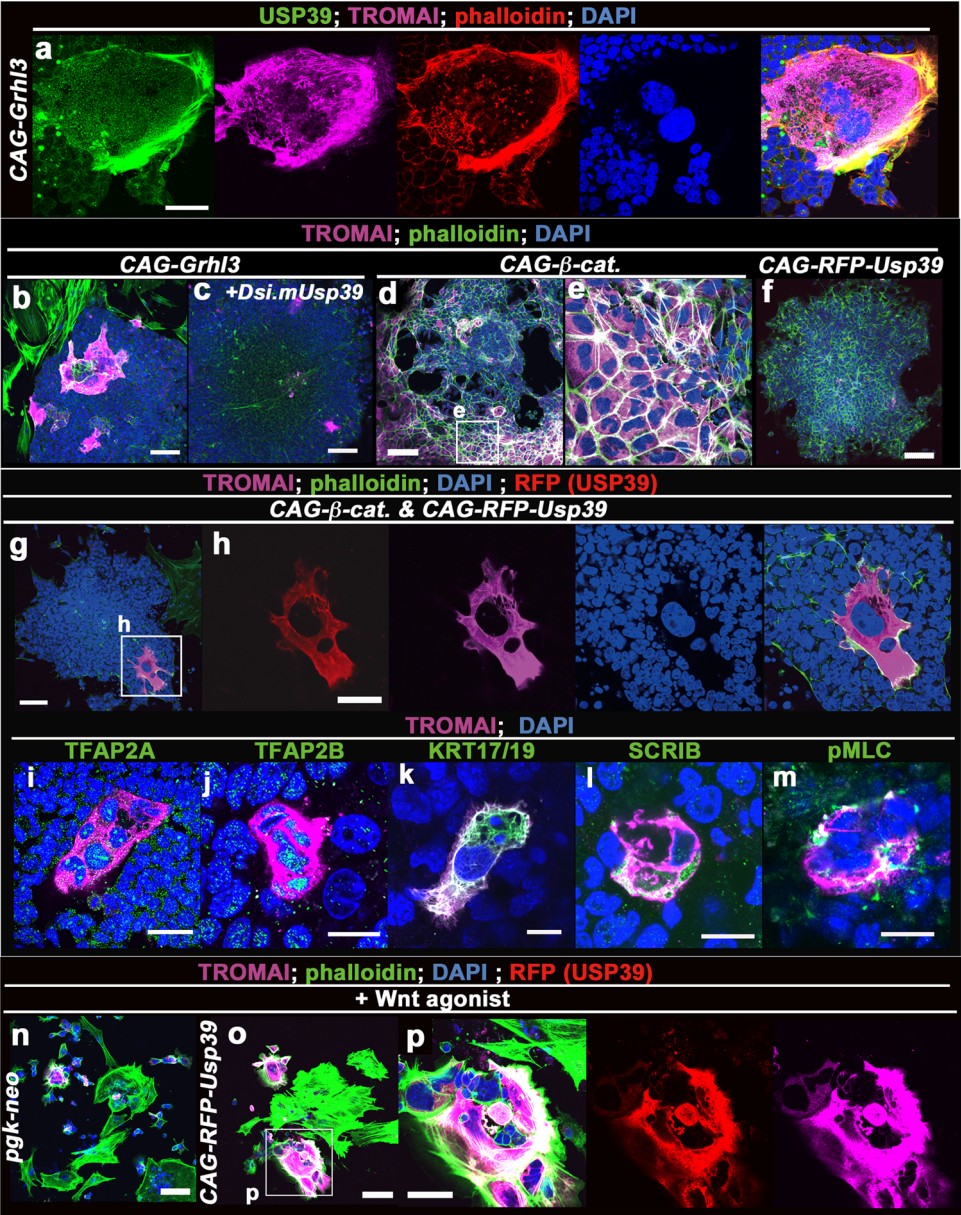

**Fig. 3 USP39 promotes formation of LM-epidermal cells by activating non-canonical Wnt signaling. a** Immunofluorescence staining for ubiquitin-specific protease 39 (USP39; green), cytokeratin 8 (TROMA-1 clone, magenta), phalloidin (F-actin, red), and 4,6′-diamidino-2-phenylindole DAPI (blue) in a large and mature (LM)-epidermal cell. USP39 protein was localized in the cytokeratin- and F-actin–enriched cytoplasmic region of the LM-epidermal cell induced by *CAG-Grhl3* from an embryoid body (EB). **b, c** Embryonic stem (ES) cells were transfected with *CAG-Grhl3* alone (**b**), and *CAG-Grhl3* and Dicer-substrate small interfering (Dsi)RNA of *Usp39* (**c**). Anti-cytokeratin 8 (TROMA-1; magenta), phalloidin (green) and DAPI (blue). EB cells transfected with *CAG-Grhl3* induce LM-epidermal cells (**b** 100% n = 30). In contrast, EB cells transfected with *CAG-Grhl3* treated with Dicer-substrate siRNA of *Usp39* decreased the rate of induction of LM-epidermal cells (**c** 70% n = 20). **d–p** Activation of the canonical Wnt pathway together with USP39 can induce LM-epidermal cells. Transfection of *CAG-β-catenin S37A* (constitutive active form of β-catenin) (**d**, **e**), *CAG–RFP-Usp39* (**f**, **o**, **p**), both *CAG-β-catenin S37A* and *CAG-RFP-Usp39* (**g–m**), and *pgk-neo* (**n**) with Wnt agonist (**n**, **o**, **p**). Molecular characterization of LM-epidermal cells induced by the co-transfection of *CAG-RFP-Usp39* and *CAG-β-catenin S37A* plasmids with immunohistochemistry (**i–m**). TFAP2A (**i**; green), TFAP2B (**j**; green), keratin (KRT)17/19 (**k** green), SCRIB (**l** green) and a phosphorylated form of non-muscle myosin light chain (pMLC)(**m** green). TFAP2A and TFAP2B, two epidermal markers, and KRT17/19, a mature epidermal marker, were expressed in the LM-epidermal cells. Immunofluorescence of SCRIB (**l**) and pMLC (**m**) two non-canonical Wnt markers localized in the cytoplasm of LM-epidermal cells. Cytokeratin-8 (TROMA-1; magenta), phalloidin (green), RFP (red), and DAPI (blue). Scale bars represent 20 μm (**j–m**), 50 μm (**a**, **h**, **i**, **p**), and 100 μm (**b–g**, **n**, **o**).

to mainly analyze the F0-8 line in this study. Homozygous *Usp39* mutant (*Usp39*$^{-/-}$) embryos did not express USP39 protein while wild-type embryos expressed it ubiquitously (Fig. 4b, b′). Notably, *Usp39*$^{-/-}$ embryos were morphologically distinct from wild-type embryos already at E6.5 (Fig. 4c–h); wild-type embryos were

stretched and had an extended lumen due to forming a primitive streak while *Usp39*$^{-/-}$ embryos were compact and failed to form a primitive streak (Fig. 4c–h).

To verify the histological observation of *Usp39*-deficient embryos, we investigated their phenotypes using several specific markers

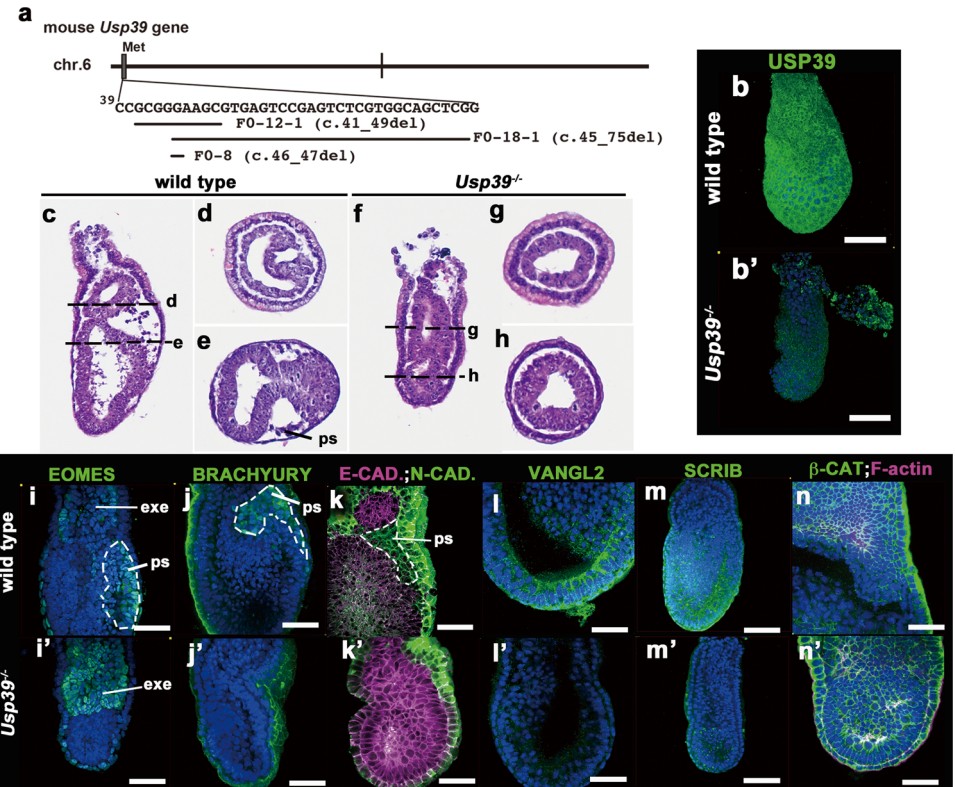

**Fig. 4 Generation and phenotype analyses of *Usp39*-deficient embryos. a** Schematic for deletion of the *Usp39* coding region generated by a CRISPR/ Cas9 system. The line F0-12-1 deletes protein coding sequences corresponding to the 41th to 49th cDNA sequence, F0-18-1 deletes the 45th to 75th cDNA sequence, and F0-8 deletes the 46th to 47th cDNA sequence, respectively. All three lines are considered to be frameshifted null alleles. **b, b′, i–n′** Immunohistochemistry of whole-mount wild-type (**b, i–n**) and *Usp39$^{-/-}$* (**b′, i′–n′**) embryos at E6.5. Ubiquitin-specific protease 39 (USP39) (green; **b, b′**), EOMESDERMIN (EOMES; green; **i, i′**), BRACHYURY (green; **j, j′**), E-/N-cadherin (E-CAD.; (magenta), N-cadherin (N-CAD.; green; **k, k′**), VANGL2 (green; **l, l′**), SCRIB (green; **m, m′**), β-catenin (β-CAT; green), F-actin (magenta; **n, n′**), and 4,6′-diamidino-2-phenylindole (DAPI) (blue). **c–h** Morphological features of wild-type (**c–e**) and *Usp39$^{-/-}$* (**f–h**) embryos at E6.5. Sagittal (**c, f**) and transverse sections (**d, e, g, h**) stained with hematoxylin–eosin. Scale bars represent 100 μm (**b, b′, c–h, m, m′**) and 50 μm (**i–l′, n, n′**).

(Figs. 4i–k′ and S5). The anterior visceral endoderm markers, *Hex*, *Lim1*, and *Fgf8* transcripts, were expressed in normal positions (the presumptive anterior region) in *Usp39$^{-/-}$* embryos at E6.5, similar to those of wild-type embryos (Fig. S5a–c′; arrowheads). An epiblast marker, *Oct3/4*, was also observed normally in *Usp39$^{-/-}$* embryos (Fig. S5d, d′). However, EOMESODERMIN and BRACHYURY proteins, two primitive streak markers, were not observed in the presumptive streak region of *Usp39$^{-/-}$* embryos, while EOMESO-DERMIN in the extra-embryonic ectoderm was present in that of *Usp39$^{-/-}$* embryos at E6.5 (Fig. 4i′, j′). To further validate defects in the formation of the primitive streak in the *Usp39$^{-/-}$* embryos, we examined the expression of E- and N-cadherins by immunohistochemistry (Fig. 4k, k′). A hallmark of epithelial–mesenchymal transition (EMT) during gastrulation, in which cells change identity from an epithelial to mesenchymal one, can be observed by a switch in cadherin expression (Fig. 4k): E-cadherin becomes down-regulated and N-cadherin becomes upregulated when epiblast cells move through the primitive streak[28] (Fig. 4k; the area surrounded by the white dashed line). In *Usp39$^{-/-}$* embryos, however, N-cadherin expression was not upregulated but E-cadherin was still expressed in the prospective EMT region (Fig. 4k′). The expression of such markers demonstrated that *Usp39* was essential to start the formation of a primitive streak correctly, but dispensable for the formation of a visceral endoderm, including anterior determination.

As shown above, considering that USP39 may activate non-canonical Wnt signaling in LM-epidermal cells in vitro (Fig. 3g–p), we examined whether molecular markers for PCP components were affected in *Usp39$^{-/-}$* embryos at E6.5 (Fig. 4l–m′). VANGL2

and SCRIB, two PCP components, were ubiquitous at the cell surface in wild-type embryos at E6.5 (Fig. 4l, m). In contrast, the expression of these two markers was markedly decreased in *Usp39$^{-/-}$* embryos (Fig. 4l′, m′), although *Vangl2* transcripts appeared to be normally expressed according to in situ hybridization (Fig. S5e, e′). However, with respect to canonical Wnt signaling, the expression of active β-catenin (α-ABC) protein, a marker of activated canonical Wnt signaling, appeared to be observed in the *Usp39*-deficient embryo, similar to that in wild-type embryos (Fig. 4n, n′). These findings together indicate that *Usp39* deficiency reduced PCP-related molecules at the protein level.

**Usp39-deficient embryos display aberrant apico-basal polarity that resembles the *Prickle1*-deficient phenotype.** To verify if USP39 is involved in PCP-dependent developmental processes, we analyzed phenotypes of *Usp39*-deficient embryos more precisely (Fig. 5). The above studies revealed that in *Usp39* null mutants, posterior patterning was severely compromised, and the primitive streak i.e. embryonic mesoderm failed to form at E6.5 (Figs. 4 and S5). Previously, a deficiency of the *Prickle1* gene, a core PCP component, also displayed failure of the primitive streak i.e. mesoderm formation[29], similar to the phenotype of *Usp39$^{-/-}$* embryos. Consistent with this observation, PRICKLE1 protein expression was down-regulated in *Usp39*-deficient embryos (Fig. 5a, b). Thus, we analyzed the apico-basal polarity phenotype of *Usp39$^{-/-}$* embryos (Fig. 5c–o). Given that *Prickle1*-deficient embryos displayed a global distortion of nuclear shape in the epiblast region, in which nuclei

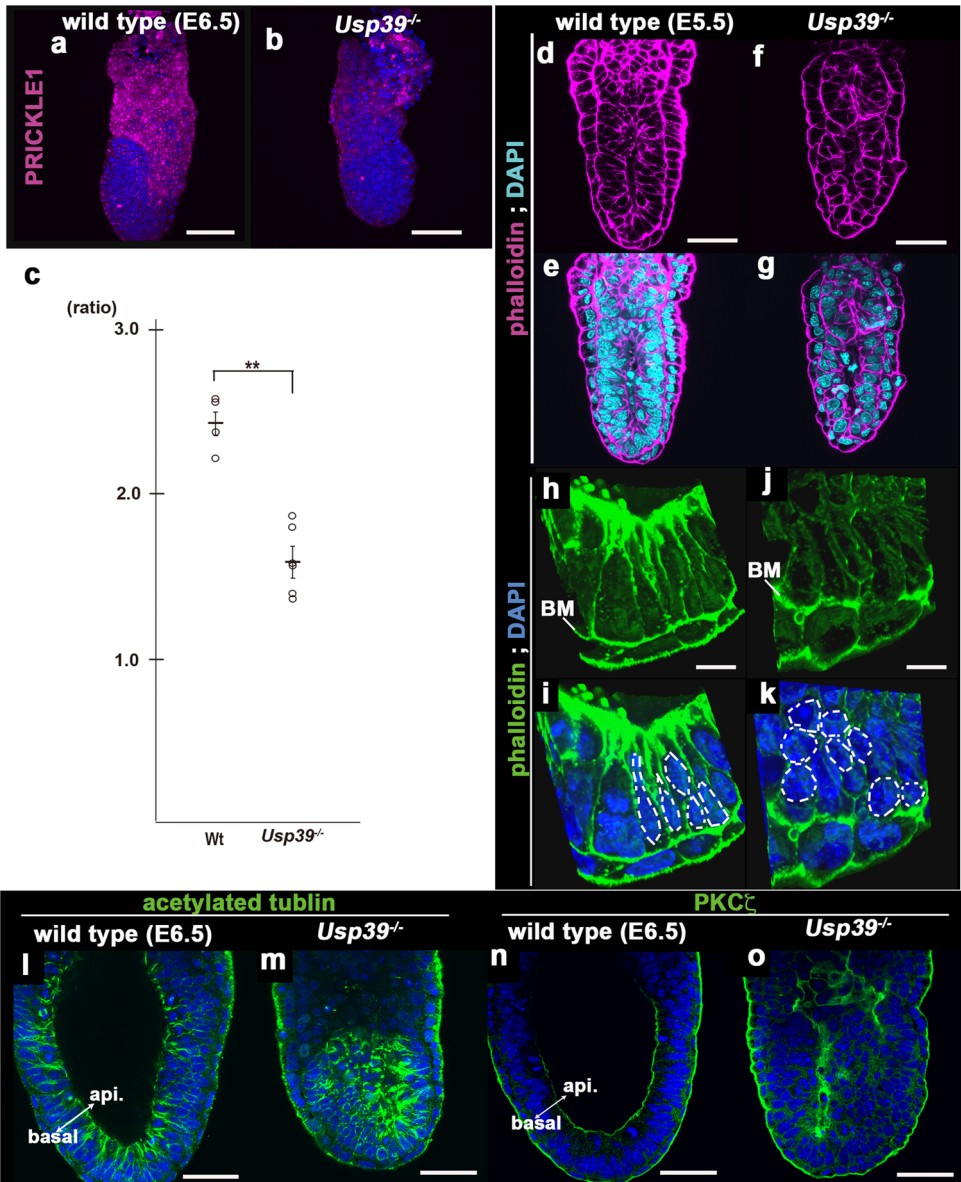

**Fig. 5 Usp39⁻/⁻ mutation embryos display aberrant apico-basal polarity in the epiblast. a, b, d–o** Immunohistochemical and fluorescence analyses. PRICKLE1 (magenta) and 4,6'-diamidino-2-phenylindole (DAPI; blue) in wild-type (**a**) and Usp39⁻/⁻ (**b**) embryos at E5.5. Phalloidin (**d–k**), acetylated-tubulin (**l, m**), protein kinase C (PKC)-ζ (**n, o**) and DAPI in wild-type (**d, e, h, i, l, n**) and Usp39⁻/⁻ (**f, g, j, k, m, o**) embryos at E5.5 (**d–g**) and E6.5 (**h–o**). Usp39⁻/⁻ epiblasts have defective cytoskeletal actin polarization and abnormal localization of acetylated tubulin and PKCζ on a sagittal orientation (**m, o**). **c** Quantification of epiblast nuclei in wild-type and Usp39⁻/⁻ embryos at E5.5. Double astersik denotes a significant difference ($p < 0.01$) compared with control (see Fig. S6). Scale bars represent 10 μm (**h–k**), 50 μm (**d–g, l, m**), and 100 μm (**a, b**).

were more spherical and randomly orientated than those of the wild type[29], we scaled the ratio of a vertical-to-horizontal length of nucleus in the epiblasts of wild-type and Usp39⁻/⁻ embryos in reference to the basement membrane (Figs. 5c–k and S6). We found that wild-type nuclei were ellipsoid and elongated along the apico-basal axis, with an average vertical-to-horizontal ratio of 2.42 (Figs. 5c and S6). In contrast, the Usp39⁻/⁻ nuclei were more spherical, with an average vertical-to-horizontal ratio of 1.58 (Figs. 5c and S6). Consistent with the observation that localization of acetylated-tubulin and PKCζ, an important factor for orientated divisions of apico-basal polarity, is also aberrant in Prickle1⁻/⁻ epiblasts[29], Usp39-deficient embryos displayed a mislocalization of acetylated tubulin and PKCζ (Fig. 5l–o). Moreover, concurrent with the evidence that conditional knockout embryos of the Prickle1 gene have no discernible effect on hair cell polarity in the cochlea[30,31],

apparent abnormalities in hair cell polarity were not evident in Usp39⁺/⁻ nor Usp39⁺/⁻; Loop tail (Lp) heterozygous (Vangl2^{Lp/+}) compound cochlea (Fig. S7a–j). Altogether, these findings together indicate that Usp39-deficient embryos show phenotypes similar to those observed in Prickle1-deficient embryos, supporting the role played by USP39 in the PCP pathway partly through PRICKLE1 expression.

**Usp39 genetically interacts with Vangl2 in axial elongation.** To explore whether the genetic interaction between Usp39 and non-canonical Wnt signaling can be observed during mouse development, we crossed Usp39⁺/⁻ and Vangl2^{Lp/+} mutant mice (Fig. 6)[32,33]. The latter carry a homolog of Drosophila starry night (also known as flamingo), a crucial PCP component, that is mutated and is a

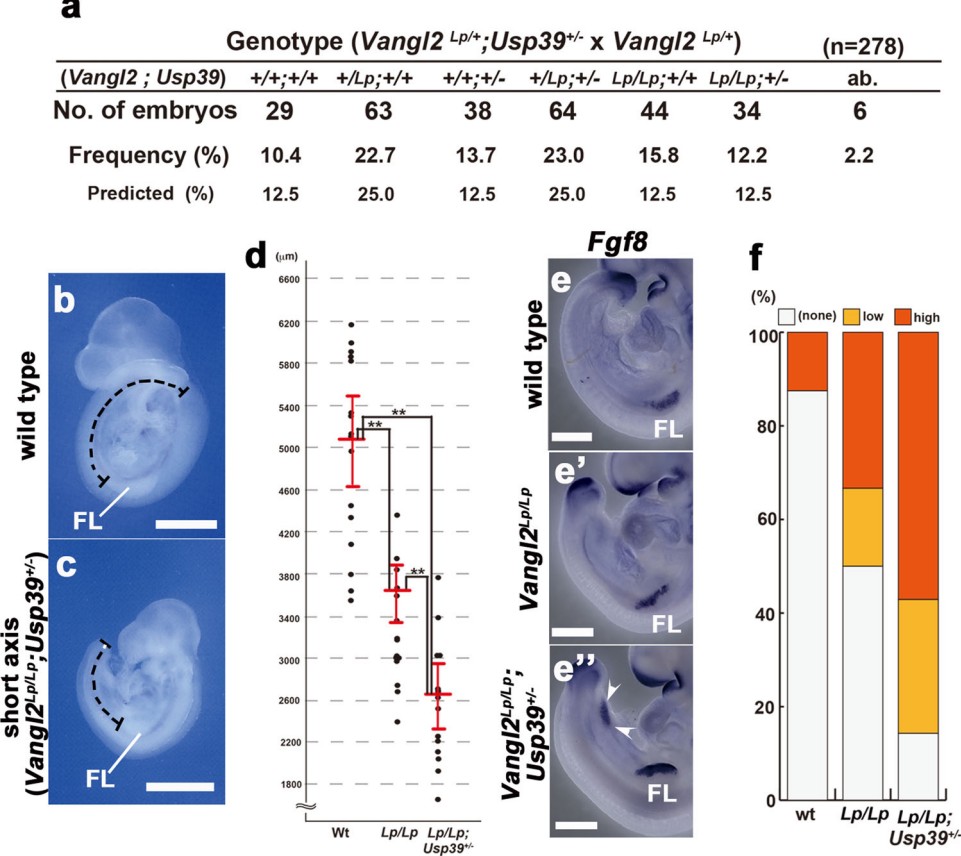

**Fig. 6 Axial elongation of $Vangl2^{Lp/Lp}$ embryos is shortened by removal of one copy of the $Usp39$ gene. a** Frequencies of genotypes among embryos from crossing $Vangl2^{Lp/+}$; $Usp39^{+/-}$ with $Vangl2^{Lp/+}$ mutant mice at E9.5. **b, c** Morphological features of wild-type (**b**) and $Vangl2^{Lp/Lp}$; $Usp39^{+/-}$ (**c**) embryos at E9.5. **d** Measured longitudinal lengths, from the ventral forelimb to caudal tip of the tail bud, in wild-type, $Vangl2^{Lp/Lp}$, and $Vangl2^{Lp/Lp}$; $Usp39^{+/-}$ embryos at E9.25 (**b, c**; dashed lines, **d**). **e–e″** In situ hybridization for $Fgf8$ expression in wild-type (**e**), $Vangl2^{Lp/Lp}$ (**e′**), and $Vangl2^{Lp/Lp}$; $Usp39^{+/-}$ (**e″**) embryos. $Fgf8$ expression in the presumptive hindlimb region is ectopically observed in $Vangl2^{Lp/Lp}$; $Usp39^{+/-}$ embryos (**e″**, white arrowheads). **f** Quantification of ectopic $Fgf8$ expression in the presumptive hindlimb region of wild-type, $Vangl2^{Lp/Lp}$, and $Vangl2^{Lp/Lp}$; $Usp39^{+/-}$ embryos at E9.25. FL forelimb. Scale bars represent 100 μm (**b, c**) and 50 μm (**e–e″**).

powerful tool for investigating the functions of non-canonical Wnt signaling since the underlying mutation only affects $Vangl2$, perturbing its trafficking to the cell membrane[34–36]. $Vangl2^{Lp/Lp}$ mice die *in utero* due to severe defects in neural tube closure. This phenotype, called craniorachischisis, is a condition in which the neural tube fails to initiate closure and remains completely open from the midbrain to tail as a fully penetrant phenotype; however, heterozygous $Vangl2^{Lp/+}$ mice survive[33]. We then generated a compound heterozygous mutant ($Usp39^{+/-}$; $Vangl2^{Lp/+}$) mouse by crossing $Usp39^{+/-}$ with $Vangl2^{Lp/+}$ (Fig. 6a). Compound heterozygous ($Usp39^{+/-}$; $Vangl2^{Lp/+}$) mice were able to survive to become fertile (Fig. 6a). By crossing $Usp39^{+/-}$; $Vangl2^{Lp/+}$ mice with $Vangl2^{Lp/+}$ mice, we obtained $Usp39^{+/-}$; $Vangl2^{Lp/Lp}$ embryos in accordance with the Mendelian ratio (Fig. 6a). At E9.25, $Vangl2^{Lp/Lp}$ homozygous embryos showed that axial extension was disrupted before the onset of neural tube closure[37]. Notably, $Usp39^{+/-}$; $Vangl2^{Lp/Lp}$ embryos exhibited a much shorter axis than $Vangl2^{Lp/Lp}$ embryos (Fig. 6b–d). To evaluate this defect more precisely, we measured the total length from forelimb to the caudal tip of the tail (Fig. 6b–d; dashed black lines). The axial length of $Usp39^{+/-}$; $Vangl2^{Lp/Lp}$ embryos was significantly shorter than that of $Vangl2^{Lp/Lp}$ littermates (Fig. 6d). In addition, $Fgf8$ expression, corresponding to the presumptive hindlimb bud, was induced in $Usp39^{+/-}$; $Vangl2^{Lp/Lp}$ embryos much more widely and intensely than that of $Vangl2^{Lp/Lp}$ and wild-type embryos (Fig. 6e–f). To confirm the precocious

induction of the hindlimb in $Usp39^{+/-}$; $Vangl2^{Lp/Lp}$ embryos, we examined $Hoxd12$ expression (Fig. S8). $Hoxd12$ transcripts were expressed in fore- and hindlimbs specifically at E10.5 under normal conditions but not earlier than E9.25 in wild-type and $Vangl2^{Lp/Lp}$ embryos (Fig. S8a–e). However, the expression of $Hoxd12$ transcripts in $Usp39^{+/-}$; $Vangl2^{Lp/Lp}$ embryos was accelerated in the hindlimb, specifically at E9.25 (Fig. S8f, g; white arrowheads)[38]. This aforementioned ectopic expression of $Fgf8$ and $Hoxd12$ is consistent with precocious hindlimb development due to the shorter axial length of $Usp39^{+/-}$; $Vangl2^{Lp/Lp}$ embryos as reported previously[38]. Taken together, these findings demonstrate that $Usp39$ genetically interacts with $Vangl2$ in terms of axial elongation, supporting the notion that USP39 contributes to modulation of non-canonical Wnt signaling.

Next, to exclude the possibility of any involvement of USP39 in canonical Wnt signaling, we tested for any genetic interaction between $β$-catenin and $Usp39$ (Fig. S9). $Usp39^{+/-}$ mutant mice were mated with mice bearing the $β$-catenin null allele; subsequently, double mutant phenotypes were examined (Fig. S9). $β$-catenin–deficient embryos show more severe phenotypes than those observed in $Usp39^{-/-}$ embryos, i.e. defects in the development of the early anterior–posterior axis prior to primitive streak formation (Fig. S9a–f). Thus, we produced $Usp39^{-/-}$; $β$-catenin$^{+/-}$ embryos and compared these to double mutant and $Usp39^{-/-}$ embryos (Fig. S9g–l). Consequently, in $Usp39^{-/-}$; $β$-catenin$^{+/-}$ embryos the anterior determination was correctly formed as well as in $Usp39$-

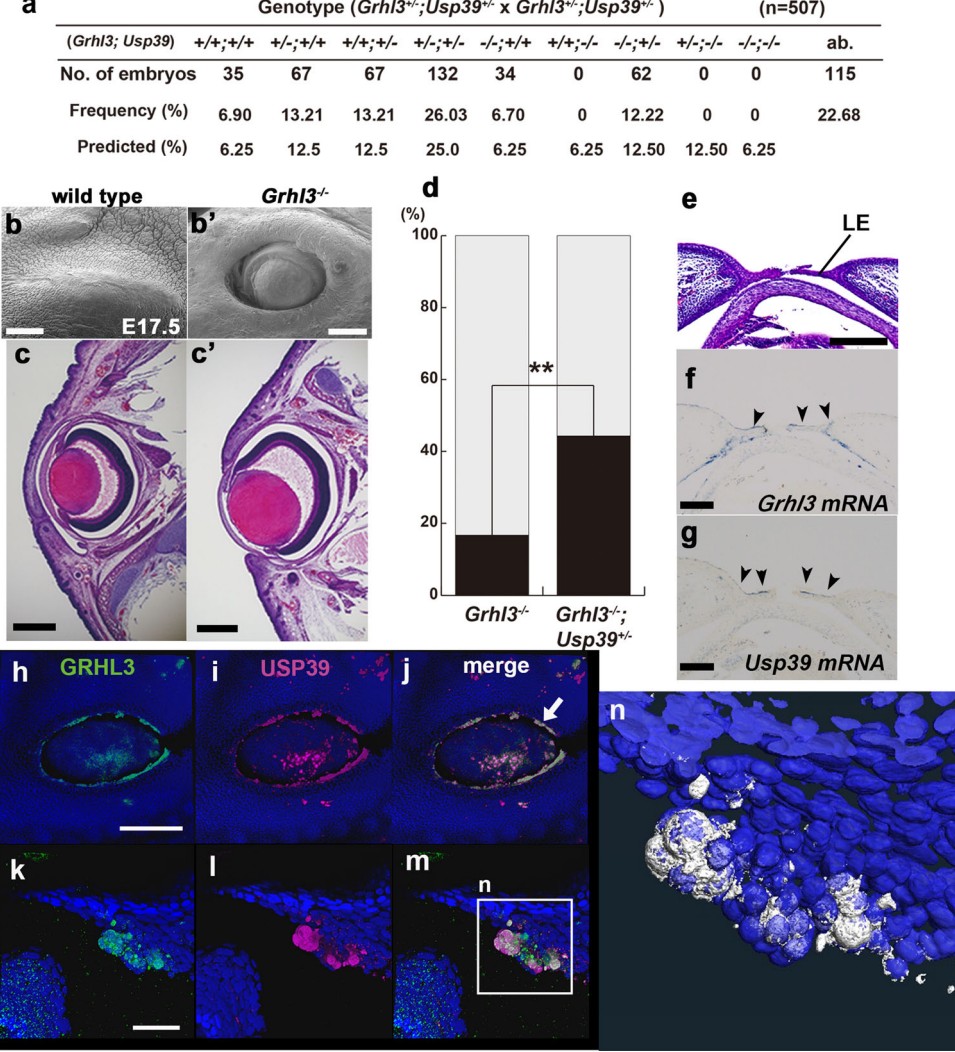

**Fig. 7 *Usp39* genetically interacts with *Grhl3* in epithelial morphogenesis during eyelid closure. a** Frequencies of genotypes among embryos from crossing *Grhl3*$^{+/-}$; *Usp39*$^{+/-}$ with *Grhl3*$^{+/-}$; *Usp39*$^{+/-}$ mutant mice at E18.5. *Usp39*$^{-/-}$ embryos, such as *Usp39*$^{-/-}$, *Usp39*$^{-/-}$; *Grhl3*$^{+/-}$, and *Usp39*$^{-/-}$; *Grhl3*$^{-/-}$ embryos, were not observed at E18.5. **b, b'** Scanning electron micrographs of E17.5 eyelids from wild-type (**b**) and *Grhl3*$^{-/-}$ (**b'**) fetuses. The eyelid is closed in the wild-type embryo at E17.5 (**b**). The eyelid is still open in the *Grhl3*$^{-/-}$ fetus at E17.5 (**b'**). **c, c'** Hematoxylin and eosin staining of an eyelid from wild-type (**c**) and *Grhl3*$^{-/-}$ (**c'**) embryos at E17.5. **d** Penetrance of the open-eye phenotype in *Grhl3*$^{-/-}$ ($N = 60$) and *Grhl3*$^{-/-}$; *Usp39*$^{+/-}$ ($N = 70$) embryos. **$p < 0.01$ ($\chi^2$ test). **e** Hematoxylin and eosin staining from a wild-type embryo at E15.5. **f, g** *Grhl3* and *Usp39* mRNA expression in the eyelid region. *Grhl3* transcripts were highly expressed in the epithelium of the leading edge at E15.5 (**f** arrowheads). *Usp39* transcripts were also detected in the dorsal leading-edge region at E15.5 (**g** arrowheads). **h–m** Merged image of Grainyhead-like 3 (GRHL3; green) and ubiquitin-specific protease 39 (USP39) proteins (magenta) demonstrating a large overlap (**j, m** flattened peridermal cells) eyelids of wild-type embryos at E15.5. **n** Reconstructed three-dimensional-confocal images of overlapped USP39 and GRHL3 protein expression [white] in the panel **m** using Amira 6.2.0 software. Scale bars represent 50 μm (**k–m**), 300 μm (**b, b'**), 500 μm (**c, c'**), and 200 μm (**e–g, h–j**).

deficient embryos (Fig. S9i, l); *Hex* and *Cer-l* transcripts, two anterior visceral endoderm markers, were detected normally in *Usp39*$^{-/-}$; *β-catenin*$^{+/-}$ embryos (Fig. S9i, l). These results indicate that no clear genetic interaction is evident between *Usp39* and *β-catenin*. These findings further support the idea that USP39 is involved in the non-canonical rather than canonical Wnt pathway.

**Usp39 interacts genetically with *Grhl3* primarily through non-canonical Wnt signaling for epithelial morphogenesis.** Since USP39 appears to contribute to the cytoplasmic localization of GRHL3 (Figs. 1, 2, and S2–4), we investigated whether *Usp39* can genetically interact with *Grhl3* during epidermal morphogenesis (Fig. 7). We crossed *Grhl3*$^{+/-}$ and *Usp39*$^{+/-}$ mice to generate compound *Grhl3*$^{+/-}$; *Usp39*$^{+/-}$ mice (Fig. 7). Developmental abnormalities were rarely observed in compound heterozygous

*Grhl3*$^{+/-}$; *Usp39*$^{+/-}$ embryos, and adult animals were fertile. However, some of the *Grhl3*$^{+/-}$; *Usp39*$^{+/-}$ double heterozygous mice displayed thickened scaly skin and failed to grow hair when compared with wild-type mice at postnatal day 14 (Fig. S10). Next, to further analyze *Grhl3*$^{-/-}$; *Usp39*$^{+/-}$ embryos, we intercrossed *Grhl3*$^{+/-}$; *Usp39*$^{+/-}$ mice to obtain these (Fig. 7a). Most of the genotypes of the offspring were transmitted from parents according to Mendelian ratios at E17.5, with the exception of three genotypes carrying the *Usp39* homozygous mutation: *Grhl3*$^{+/+}$; *Usp39*$^{-/-}$, *Grhl3*$^{+/-}$; *Usp39*$^{-/-}$, and *Grhl3*$^{-/-}$; *Usp39*$^{-/-}$ (Fig. 7a). To evaluate the genetic interaction between *Grhl3* and *Usp39*, we analyzed phenotypes of epidermal development during eyelid closure since *Grhl3*$^{-/-}$ embryos have been reported to be defective in eyelid closure with a low penetrance (Figs. 7b–c' and S11a-i)[39,40]. Morphological examination of

eyelids revealed that the eyelid-opened phenotype was much more frequently observed in $Usp39^{+/-}$; $Grhl3^{-/-}$ mutant embryos ($n = 31/70$; 44.28%) than in $Grhl3^{-/-}$ embryos ($n = 10/60$; 16.67%) at E17.5 (Fig. 7d). To examine anatomical features in further detail, we employed scanning electron microscopy (Fig. S11a–i). Typically, at E14.5, wild-type, $Grhl3^{-/-}$, and $Grhl3^{-/-}$; $Usp39^{+/-}$ embryos manifested opened eyes of similar shape and showed no distinct morphological differences. The morphological differences between wild-type and mutant embryos were evident at E15.5 (Fig. S11a–i). In the wild-type embryo, the rims of the top and bottom eyelids extended toward each other and were fusing from both nasal and lateral margins (Fig. S11a–c). In contrast, the eyelids of $Grhl3^{-/-}$ and $Grhl3^{-/-}$; $Usp39^{+/-}$ embryos failed to extend during this period, leaving the ocular surface exposed (Fig. S11d–i). These findings indicate that $Grhl3$ is able to interact with $Usp39$ genetically in terms of eyelid closure.

To explore the exact timing and place of GRHL3 interaction with the USP39 protein in epidermal cells, we analyzed expression of the two genes and eyelid phenotypes more closely (Fig. 7e–n). In wild-type mice, eyelids grow across the eye and fuse together around E15.5; embryos are born with their eyes closed. The processes of embryonic eyelid development are described as follows[41,42]: initial ectodermal morphogenesis and groove formation (E11.5), eyelid mesenchymal protrusion (E13.5), protruding epithelial ridge formation at the tip of the eyelid margin (E15.5), subsequent extension of the upper and lower eyelid epithelium (E15 to E16), and consequent extension of mesenchymal cells (E16.5 to E17.5)[41,42]. The fusion of upper and lower eyelids occurs only in the epithelial layer, but mesenchymal cells are not fused for subsequent reopening until around 2 weeks postnatally. To determine when $Grhl3$ interacts with $Usp39$ during eyelid closure, we examined $Grhl3$ and $Usp39$ expression at the mRNA level in sections, and at the protein level in whole-mount of wild-type embryos at E15.5 (Figs. 7e–g and S11r–s''). In addition, $Grhl3$-$\beta$-$gal$ expression, which is marked by $Cre$-$IRES$-$NLS$ $lacZ$ driven by the $Grhl3$ locus, was utilized to detect keratinocytes at the outer layer of the migrating leading edge (flattened periderm cells) at E15.5 as reported previously[39]. The expression patterns of GRHL3 protein as well as those of its transcripts correlated with a region of dense cell packing (Fig. S11p–q''). Further precise expression analysis of the subcellular localization of two proteins with their specific antibodies revealed that USP39 and GRHL3 proteins appeared to overlap with the cytoplasm of the migrating apical cells to close the eyelids (Figs. 7h–n and S11r–s''). Since $Grhl3$ is required for the formation of normal actin bundles during the development of a leading edge[39], we examined F-actin expression in wild-type embryos during eyelid closure (Fig. S11j, k). We found that F-actin expression became progressively stronger and converged at the leading edge, which may provide the mechanical force required for the formation of filopodia and cell migration of wild-type embryos at E15.5 (Fig. S11j, k). In contrast, in $Grhl3^{-/-}$ and $Grhl3^{-/-}$; $Usp39^{+/-}$ eyelids, F-actin formation was greatly reduced and did not converge at the leading edge (Fig. S11l–o). These results demonstrated that $Grhl3$ was involved in actin polymerization at the leading edge of epidermal cells during eyelid closure. The presence of such phenotypes is in agreement with our finding that USP39 can activate PCP components interacting with GRHL3 protein and enhance actomyosin networks in LM-epidermal cells in vitro (Fig. 3). Accordingly, $Usp39$ transcripts and protein were also highly expressed at the leading edge of the eyelid where these overlapped with GRHL3-expressing cells (Figs. 7e–n and S11p–s''). These findings support the notion that GRHL3 and USP39 have cooperative roles in the eyelid closure process.

In order to verify whether USP39 and GRHL3 proteins are in very close proximity, we performed a proximity ligation assay (PLA), also referred to as Duolink PLA technology, that permits the detection of protein–protein interaction in situ (at distances <40 nm) at endogenous protein levels (Fig. 8e)[43,44]. The PLA exploits specific antibodies identifying the two proteins of interest and takes advantage of specific DNA primers covalently linked to the antibodies. A hybridization step followed by PCR amplification with fluorescent probes then permits visualization of spots of proximity by fluorescence microscopy with high sensitivity and specificity (Fig. 8e). Thus, in situ PLA is an advanced method to detect protein interactions in cells or tissues as long as suitable antibodies are available. Primary antibodies for detecting the desired protein–protein interaction were derived from two different species: rabbit (anti-USP39) and mouse (anti-GRHL3; Figs. 7h–n and 8a–d). Notably, immunohistochemical studies indicated that considerable amounts of cytoplasmic GRHL3 appeared to co-localize with USP39 in MBT2 cells as well as periderm cells during eyelid closure (Figs. 7j, m, n and 8c, d). Furthermore, confocal images of in situ PLA using rabbit anti-USP39 and mouse anti-GRHL3 antibodies in MBT2 and periderm cells during mouse eyelid closure revealed that USP39 was localized in close proximity to GRHL3 within 40 nm (Fig. 8f–k). These results support the notion that GRHL3 can interact with USP39 under physiological conditions.

Next, to evaluate if the above eyelid closure in $Grhl3^{-/-}$; $Usp39^{+/-}$ embryos is mediated by non-canonical Wnt signaling, we examined molecular markers for the PCP components, VANGL2 and CELSERI, and the phosphorylated form of myosin light chain (pMLC) in eyelid epithelium at E15.5 (Fig. 8l–n'). In wild-type embryos at around E15, these three PCP components were consistently observed in the periderm of the fusion line (Fig. 8l–n). In $Grhl3^{-/-}$; $Usp39^{+/-}$ mutant eyelids, however, the expression of these three markers was severely downregulated (Fig. 8l'–n'). These expression studies indicate that $Grhl3$ together with $Usp39$ contribute to eyelid closure by enhancing actomyosin networks after activating non-canonical Wnt signaling.

To further directly determine whether non-canonical Wnt signaling can contribute to defects in eyelid closure due to reduced dosage of the $Usp39$ gene, we cultured explants of mouse eyelid anlagen in the presence of the chemical inhibitors, Y27632 (ROCK inhibitor) and NSC23766 (Rac inhibitor) (Fig. S12)[45,46]. Most eyelid explants from wild-type mouse embryos at E15.25 can be closed within 24 h of culture (Fig. S12a, b, i). The epithelial extension, which was observed in the periderm of fusion lines, was replicated in the wild-type explants (Fig. S12c–h). Notably, treatment with Y27632 and NSC23766 hardly affected the closure of eyelids in wild-type explants (Fig. S12i). Since $Vangl2^{Lp/Lp}$ embryos display eyelid closure defects with 100% penetrance, we exploited $Vangl2^{Lp/+}$ eyelids in explant culture. Indeed, in $Vangl2^{Lp/+}$ explants, defects in eyelid closure were consistently found; the opened-eye phenotype was 25% (Fig. S12i). Additionally, Y27632 or NSC23766 treatment led to an increase in the opened-eye phenotype by up to more than 60% and 40%, respectively (Fig. S12i). These findings support the idea that non-canonical Wnt signaling is involved in eyelid closure. Moreover, in $Usp39^{+/-}$ and $Vangl2^{Lp/+}$; $Usp39^{+/-}$ explants, Y27632 or NSC23766 treatment significantly increased the opened-eye phenotype (Fig. S12i). These findings together demonstrate that non-canonical Wnt signaling contributes to eyelid closure in cooperation with $Usp39$ in this context.

To verify whether $Usp39$ genetically interacts with $Grhl3$ through the PCP but not canonical Wnt pathway, we generated a $Usp39$ and $Grhl3$ $^{NLS}$ double mutant mouse, in which GRHL3 localized to the nucleus rather than the cytoplasm, by inserting a nuclear localization signal (NLS) sequence into the translational

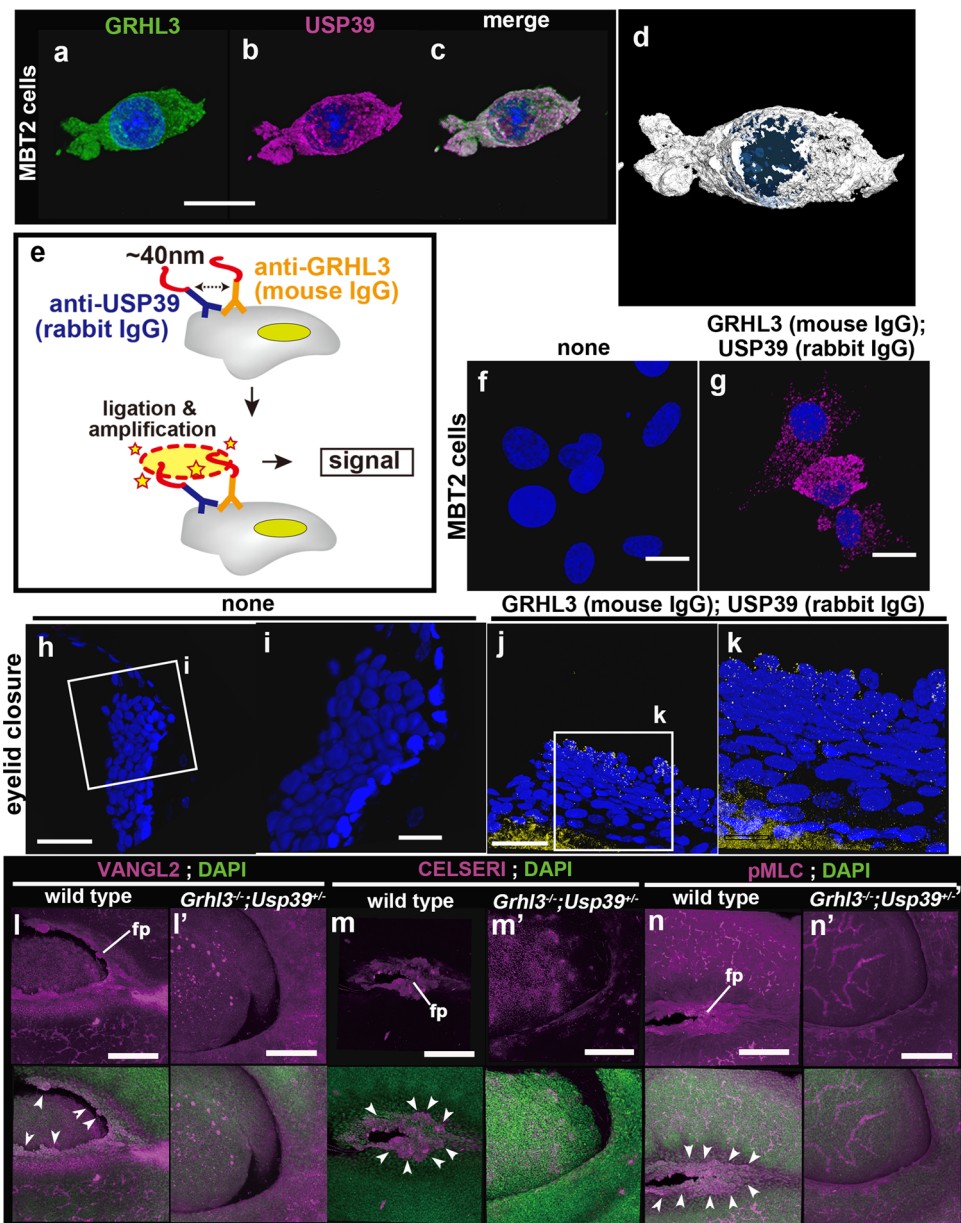

**Fig. 8 Detection of USP39-GRHL3 interactions in situ using a Duolink proximity ligation assay (PLA) at endogenous protein levels and expression of planar cell polarity (PCP)-related molecules in *Grhl3*⁻/⁻; *Usp39*⁺/⁻ mutant eyelids. a–d** Co-localization between GRHL3 (green) and USP39 (magenta) in MBT2 cells by immunohistochemistry and analyzed with Amira software (**d**). **e** Schematic representation of PLA technology. **f–k** The specific interactions between USP39 and GRHL3 proteins can be detected as PLA signals (**g**, **j**, **k**) in MBT2 cells (**f**, **g** magenta) and eyelid tissues (**h–k** yellow). Technical negative controls are conducted in the absence of the first antibodies (none) (**f**, **h**, **i**). **l–n′** Planar cell polarity–related molecules are greatly reduced in *Grhl3*⁻/⁻; *Usp39*⁺/⁻ mutant eyelids. Arrowheads indicate the corresponding expression in the wild-type flattened periderm cells (magenta). Expression of VANGL2 (**l**, **l′**), CELSERI (**m**, **m′**), and phosphorylated myosin light chain (pMLC) (**n**, **n′**) in wild-type (**l–n**) and *Grhl3*⁻/⁻; *Usp39*⁺/⁻ (**l′–n′**) embryos. fp flattened peridermal cells. Scale bars represent 20 μm (**a–d**, **f**, **g**, **i**, **k**), 50 μm (**h**, **j**) and 200 μm (**l–n′**).

start site of the *Grhl3* locus (Fig. S13a)[9]. *Grhl3*^NLS/NLS^ defects are due to failure of cytoplasmic GRHL3-dependent epithelial morphogenesis involving the non-canonical Wnt pathway, but not nuclear GRHL3-dependent transcription involving the canonical Wnt pathway[9]. NLS-fused GRHL3 was mainly localized in the nucleus of surface ectoderm cells in *Grhl3*^NLS/NLS^ embryos as compared with normal full-length GRHL3 in the wild type at E8.5 (Fig. S13b–e). *Grhl3*^NLS/NLS^ mutant embryos display neural tube closure defects and their severity is milder than that of the *Grhl3*-deficient embryos[9]. To evaluate the severity of defects in the double mutant more quantitatively, we measured the degree of

protruding neural tissues considered the sacral spina bifida phenotype (Fig. S13f–h). We generated *Grhl3*^NLS/NLS^; *Usp39*⁺/⁻ compound mutants and measured the degree of protruding neural tube. The penetrance of sacral spina bifida was almost similar between *Grhl3*^NLS/NLS^ and *Grhl3*^NLS/NLS^; *Usp39*⁺/⁻ embryos ($n = 7/11$ and $n = 16/24$, respectively; $p = 0.860 > 0.05$). However, *Grhl3*^NLS/NLS^; *Usp39*⁺/⁻ embryos appeared to display a much more severe spina bifida phenotype than *Grhl3*^NLS/NLS^ embryos (Fig. S13g, h). These findings together further support the concept that USP39 contributes to GRHL3-dependent developmental processes through the PCP but not canonical Wnt pathway.

**USP39 regulates migration of mesendoderm cells by upregulation of PRICKLE1 expression through its deubiquitination activity**. Although *Usp39* is involved in RNA splicing[23,24] and also functions as a deubiquitinating enzyme[25], the molecular mechanism underlying how USP39 regulates the non-canonical Wnt pathway is still unknown. First, to test whether aberrant RNA splicing may affect *Usp39* mutant phenotypes, we analyzed mRNA expression profiles of *Usp39*-deficient and wild-type embryos by conducting RNA sequencing (Supplementary Data 1 and 2). Although *Vangl2* expression in *Usp39*-deficient embryos was apparently reduced at the level of protein but not of mRNA [fold change = −1.0731, *p*-value < 0.05] (Figs. 4l, l′ and S5e′), no aberrant splicing was found in *Usp39*-deficient *Vangl2* transcripts. In addition, *Rb1* transcripts displayed splicing defects because of USP39 mutation in zebrafish;[47] such a splicing abnormality appeared to be undetectable in *Usp39*-deficient embryos. Consequently, comparative expression profiles revealed that four transcripts of PCP-related genes, *Dact1*, *Ror2*, *Prickle1*, and *Fzd2*, were significantly down-regulated in *Usp39*−/− compared to wild-type embryos at E6.5 (fold change >2, and *p*-value < 0.05; Fig. S14a and Supplementary Data 1 and 2). However, all four transcripts of these PCP genes did not show aberrant splicing patterns. These results suggest that *Usp39*-deficient phenotypes are not due to the aberrant splicing of PCP genes.

Next, we hypothesized that USP39 may function as a deubiquitinating enzyme for primarily stabilizing PCP components. To test this possibility, we generated *Ube1* mutant mice using CRISPR/Cas9 genome editing and produced *Ube1;Usp39* double mutant mice (Fig. S14c). Ubiquitin-activating enzyme E1 (UBE1) catalyzes the first step of the ubiquitin–protease system in mammals (Fig. S14b). Since *Ube1* is located on the X chromosome, male *Ube1* mutant (*Ube1*⁻) embryos displayed reduced UBE1 expression (Fig. S14d, e) and different morphological features from those of wild-type embryos at the blastocyst stage (Fig. S14f–g′). In order to examine if the UBE1 factor is involved in polyubiquitination in early mouse embryos, we analyzed expression studies by immunohistochemistry using an anti–poly-Ub antibody at E4.0 (Fig. S14h, i) and found that male *Ube1*⁻ mutant embryos showed reduced polyubiquitination compared to wild-type embryos (Fig. S14h, i). In hematoxylin–eosin-stained sections of whole decidual swelling at E5.5 from female *Ube1* heterozygous mice crossed with male wild-type mice, the resorbed structure was consistently observed in approximately a quarter of offspring (Fig. S14l, m). These findings indicate that *Ube1* deficiency causes peri-implantation lethality, supporting a crucial role for *Ube1* during early embryogenesis. Then, we examined phenotypes of *Usp39*−/−; *Ube1*+/− female embryos by crossing *Usp39*+/−; *Ube1*+/− female mice with *Usp39*+/−; *Ube1*+ male mice (Figs. 9a–l and S14n). Marker analysis revealed that in *Usp39*−/− embryos the primitive streak was not formed yet at E6.5 (Figs. 4, 5, and 9b) but mesendoderm was partially formed at around E7.5 (Fig. 9e, h, k); i.e. mesendoderm cells marked by EOMESODERMIN, *Hex*, and *Foxa2* expression were able to be formed belatedly but were unable to migrate toward the anterior side and remained on the posterior side (Fig. 9e, h, k). In contrast, in *Usp39*−/−; *Ube1*+/− embryos, the primitive streak formed partially at E6.75 (Fig. 9c) and mesendoderm cells appeared to migrate toward the anterior side compared to *Usp39*−/− embryos at E7.5 (Fig. 9f, i, l). These findings demonstrate that failure in *Usp39*−/− mesendoderm migration can be partially restored by removal of one copy of the *Ube1* gene, i.e. probably due to the reduction in the ubiquitination activity. These findings suggest that USP39 contributes to mesendoderm migration via its deubiquitination activity. To further support this hypothesis, we

tested if USP39 activity could control expression of PCP components such as PRICKLE1 in mesendoderm cells and found that PRICKLE1 protein was expressed in mesendoderm cells (Fig. 9m, n). Moreover, PRICKLE1 expression appeared to be reduced in *USP39* knockdown cells similar to that of *Usp39*-deficient embryos (Figs. 9o–r and 5b). Taken together, the above lines of evidence support the notion that USP39 may participate in the upregulation of PCP components partly through its deubiquitination activity in mammalian morphogenesis.

## Discussion

Current studies, albeit highly reliant on over-expression, indicate that USP39 can interact with the Ub-like domain of GRHL3 for proper localization of GRHL3 in the cytoplasm (Figs. 1–3 and S2–4). Specifically, USP39 and GRHL3, at physiological concentrations, co-localized in close proximity (within 40 nm) in MCF7 and periderm cells during eyelid closure (Fig. 8f–k). However, further biochemical studies will be necessary to elucidate whether the physical interaction between the two proteins is direct and they are present in the same complex in these cells. We have previously shown that the Ub-like domain of GRHL3 is necessary for the efficient formation of LM-epidermal cells from EBs and that this epithelial morphogenesis, such as a change in cell shape, are mediated by activation of actomyosin networks through upregulation of PCP components[9]. Thus, current results together with our previous findings support the view that the Ub-like domain of GRHL3 is crucial to its cytoplasmic function of being involved in epithelial morphogenesis via non-canonical Wnt signaling. In agreement with this hypothesis, USP39 is capable of activating non-canonical Wnt signaling during in vitro epidermal differentiation from EB cells (Fig. 3). Therefore, we propose that USP39 can activate non-canonical Wnt signaling partly by allowing GRHL3 to localize in the cytoplasm.

This study demonstrates that *Usp39* and *Grhl3* cooperate to contribute to epithelial morphogenesis during mouse eyelid closure (Figs. 7, 8, S11, and S12). Mouse *Grhl3* functions as a key regulator of the PCP signaling pathway, acting through the *RhoA* activator, *RhoGEF19*, at the level of transcription in the nucleus as well as through PCP components such as VANGL2 and RHOA at the protein level in the cytoplasm during closure of the neural tube[9]. Thus, *Grhl3*-mediated processes appear to contribute to spatiotemporal-specific remodeling of the actin cytoskeleton during epithelial morphogenesis, such as in collective cell migration, neural tube defects, and disordered cochlear polarity[9,48,49]. Notably, cytoplasmic expression of *Grhl3* is concordant with that of *Usp39* during eyelid closure, supporting the idea that gene dosages of *Grhl3* and *Usp39* are critical determinants of epithelial morphogenesis in this process (Figs. 7, 8, and S11). Consistent with a previous report describing how mice carrying mutations in *Vangl2*[32], *Scrb1*[50], *Celsr1*[51], *PTK7*[52], and *Grhl3*[40] genes exhibit an open-eyes phenotype at birth, the expression of CELSR1 and VANGL2 products was lost in *Grhl3*−/−; *Usp39*+/− mutant eyelids (Figs. 7, 8, and S11).

In addition, a *Usp39* mutation can exacerbate neural tube defects observed in a nuclear-localized mutation of *Grhl3*, i.e. the *Grhl3*NLS allele (Fig. S13). We have previously shown that nuclear-localized GRHL3 contributes to transcriptional activation of epidermal-specific genes in the canonical Wnt signaling pathway while cytoplasmic-localized GRHL3 contributes to activation of non-canonical Wnt signaling[9]. Accordingly, GRHL3 fused with the NLS sequence is able to play a role as a transcriptional activator of epidermal differentiation in the nucleus but not as an activator of PCP components in the cytoplasm (Fig. S13a)[9]. Given that *Usp39* can genetically interact with the mutation of *Grhl3*NLS allele (Fig. S13), USP39 may mainly affect

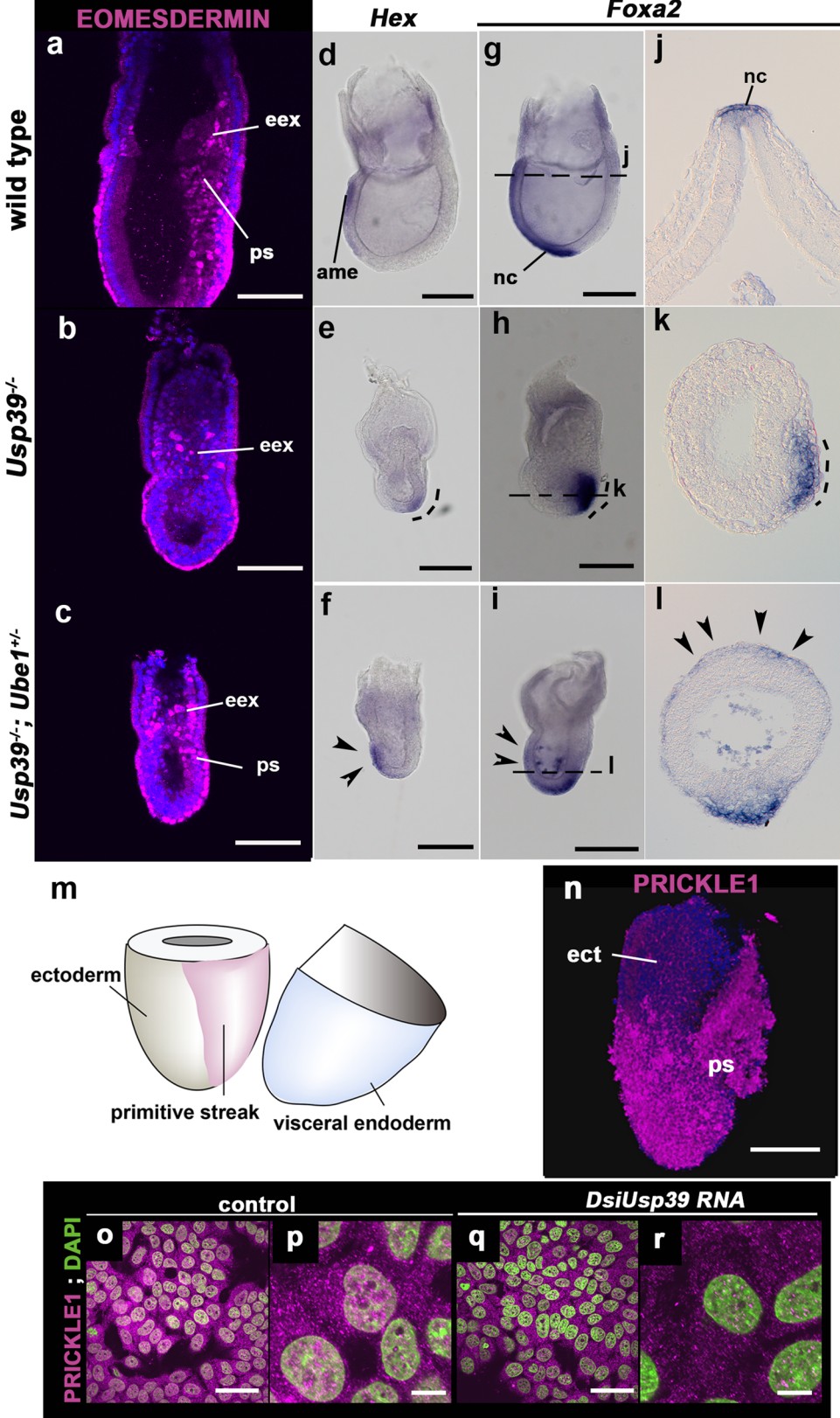

the cytoplasmic function of GRHL3, which is mediated by a non-canonical Wnt signaling pathway rather than *β-catenin*-dependent canonical Wnt pathway.

USP39 is necessary to non-canonical Wnt-dependent processes through the correct expression of PCP components during mouse epithelial morphogenesis (Figs. 4–8, 10a, and S13). PCP components play essential roles in various developmental processes that need cell polarity, such as gastrulation, inner ear hair cell patterning, neurite outgrowth, and neural tube closure, as demonstrated by genetic studies in animal models[33,51,53–57] (Fig. 10a). Here, we demonstrated that *Usp39*-deficient embryos displayed a failure in the formation of the primitive streak,

**Fig. 9 Anterior migration defects of mesendoderm cells of *Usp39* mutant embryos are restored by removal of one copy of the *Ube1* gene. a–c** Immunohistochemistry of EOMESDERMIN (extra-embryonic ectoderm, visceral endoderm, and primitive streak marker) at E6.75 of wild-type (**a**), *Usp39*$^{-/-}$ (**b**), and *Usp39*$^{-/-}$;*Ube1*$^{+/-}$ (**c**) embryos. **d–l** Whole-mount in situ hybridization of *Hex* and *Foxa2* in wild-type (d,g,j), *Usp39*$^{-/-}$ (**e, h, k**), and *Usp39*$^{-/-}$; *Ube1*$^{+/-}$ (**f, i, l**) embryos, and corresponding sagittal sections (**j–l**) are shown at E7.5. In the *Usp39*$^{-/-}$ embryos, *Hex* and *Foxa2* positive mesendoderm cells remained on the distal side (**e, h, k** dotted curved lines). The anterior movement of mesendoderm cells was partially restored in *Usp39*$^{-/-}$; *Ube1*$^{+/-}$ embryos (**f, i, l**; arrowheads). **m** A schematic illustration of ectoderm and primitive streak regions after removal of the visceral endoderm layer at E6.5. **n** PRICKLE1 protein was detected in the migrating primitive streak (ps), rather than in ectoderm (ect) at E6.5. PRICKLE1 (magenta) and 4,6′-diamidino-2-phenylindole (DAPI; blue). **o–r** PRICKLE1 protein was downregulated by *Dicer-substrate* small interfering (Dsi) RNA of *Usp39* in MCF7 cells. PRICKLE1 (magenta) and DAPI (green). ame anterior mesendoderm, ect ectoderm, eex extra-embryonic ectoderm, nc notochord, ps primitive streak. Scale bars represent 10 μm (**p**, **r**), 50 μm (**a–c**, **o**, **q**) and 100 μm (**n**), and 200 μm (**d–i**).

migration of mesendoderm cells, and apico-basal polarity of the epiblast while apparent abnormalities in the hair cell polarity of cochlea were not evident in *Usp39*-deficient embryos (Figs. 4–8, S6, S7, and S13). These *Usp39*-deficient phenotypes appear to resemble those due to *Prickle1*-deficiency[29–31]. Therefore, these findings together suggest that USP39 is necessary for non-canonical Wnt signaling-dependent developmental processes. Concurrent with the *Usp39*-deficient phenotypes, the expression of several crucial PCP components, such as VANGL2, SCRIB, PRICKLE1, and CELSER1 products, were severely reduced in *Usp39*-deficient embryos (Figs. 4, 5, and 8). Moreover, the *Usp39* mutation genetically interacts with the *Vangl2*$^{Lp}$ mutation; *Vangl2*$^{Lp/Lp}$; *Usp39*$^{+/-}$ double mutant embryos displayed shorter tails than those of *Vangl2*$^{Lp/Lp}$ embryos (Fig. 6). Core components of conserved PCP proteins, including *Vang/Stbm*, *Fmi/Stan/Celsr*, *Fz/Fzd*, *Dsh/Dvl*, *Prickle*, and *Diengo/Diversin*, play central roles in epithelial polarity formation[58–60] with their mutations displaying severe neural tube defects and a shortened anterior–posterior axis[32,51,56,61–63]. Specifically, the failure in axis elongation may reflect defective convergent extension movements, a process whereby axial mesoderm cells intercalate and converge along with the anterior–posterior axis[64]. Thus, the shorter tail phenotype in *Vangl2*$^{Lp/Lp}$; *Usp39*$^{+/-}$ mutant embryos might imply a crucial role of *Usp39* in the migration of mesendoderm cells mediated by PCP components[65].

The current study suggests that USP39 participates in the correct expression of PCP components partly through its deubiquinating enzyme activity (Figs. 9, 10b, and S14). USP39 is involved in two major biochemical functions: mRNA splicing and deubiquitination[23–25]. Here, RNA-seq analysis revealed that transcripts of several PCP components affected in *Usp39*-deficient embryos had normal splicing patterns. In contrast, a failure in migrating mesendoderm because of *Usp39*-deficiency was partly restored by *Ube1*-deficiency (Fig. 9). PCP components such as PRICKLE1 have been shown to regulate gastrulation movements[66]. These findings lead us to propose that the abnormality in mesoderm migration might be due to the loss of PCP components including PRICKLE1 protein by promoting its ubiquitination in the absence of USP39 products (*Usp39*$^{-/-}$), and the migration defect might be complemented due to the reduction in the global ubiquitination activity by the removal of one copy of the *Ube1* gene. Regulation of protein ubiquitination is considered to affect signaling activity by controlling levels of protein expression[67–71]. Notably, Smurfs, involved in protein degradation as ubiquitin ligases, play important roles in the polarized localization of PCP proteins[68]. Deubiquitinase USP29 promotes cell migration by stabilizing Snail protein[72]. A recent report describes how USP39 is able to bind nuclear-localized STAT1 and sustain the STAT1 protein level by deubiquitination; consequently, this is a means to export deubiquitinated STAT1 from the nucleus to the cytoplasm by avoiding its degradation[25]. Therefore, these lines of evidence support that USP39 controls non-canonical Wnt signaling-

dependent morphogenetic processes primarily via positively modulating expression levels of PCP components (Fig. 10b).

## Methods

**Epidermal differentiation from ES cells in vitro.** Epidermal cells were differentiated from G4 embryonic stem cells[73] via embryoid body (EB) cells[74]. The frequency of LM-epidermal cells (%) was calculated from the number of EBs having LM-epidermal cells among the total number of EBs analyzed. The plasmids, *CAG-Grhl3*, *CAG-β-catenin S37A*, *CAG-Usp39*, and *CAG-RFP-Usp39* were constructed according to standard procedures[9] (detailed procedures for plasmid constructions are available upon request). These plasmids were transfected into embryonic stem (ES) cells using Lipofectamine LTX (Invitrogen). Then, ES cells were cultured with G418 (150 μg mL$^{-1}$) for 24 h. The following chemical reagents were added to EB cells after transfection of plasmids: Wnt agonist (canonical Wnt activator; Merck; cat. no.681665; 10 μM).

**Expression analysis of GRHL3 and USP39 in cells.** Transfected plasmids, *CAG-Grhl3 full-length-EGFP*, *TA-EGFP*, *CP2-EGFP*, and *Ub-like-EGFP*, were made as previously described[9]. The open reading frame clone of *HaloTag-Usp39* from Kazusa DNA Research Institute (pFN21AE5640 or FHC28713), labeled with HaloTag TMR ligand, was made according to the manufacturer's instructions (Promega). MCF7 cells were maintained in Minimum Essential Medium (Gibco, Invitrogen) supplemented with 10% fetal bovine serum (Gibco, Invitrogen), 0.1 mM non-essential amino acid solution, and 1 mM sodium pyruvate at 37ºC in a humidified incubator with 5% $CO_2$. RT4 cells were cultured in McCoy's 5a (GE Healthcare: SH30200-01) supplemented with 10% fetal bovine serum (Gibco, Invitrogen), and 1X Glutamax (Gibco: 35050-061). MBT2 cells were cultured in Minimum Essential Media (Gibco, Invitrogen) supplemented with 10% fetal bovine serum (Gibco, Invitrogen), and 1X Glutamax. Sources of cell lines are indicated in Table S1.

Transfections of a total of 2.5 μg of a plasmid preparation for each 6-well culture were performed with Lipofectamine LTX reagent (Invitrogen) according to the instructions of the manufacturer. Transfections of *Grhl3* siRNA (ID33752 and ID33754; Ambion) and *Usp39* Dicer-substrate siRNA (hUSP39.13.1, hUSP39.13.2, and mUSP39.13.2; IDT) were performed using Lipofectamine 3000 (Invitrogen) according to the manufacturer's instructions.

**Immunohistochemistry.** Information on primary antibodies and their concentrations used in this study is outlined in Table S1. Conjugation of anti-GRHL3 (aa 478-493) with Alexa Fluor 488 dye was performed using a Alexa® 488 Microscale protein labeling kit from molecular Probes (cat. no. A30006) according to the manufacturer's instructions. For whole-mount preparations, samples were fixed in 4% paraformaldehyde (PFA) for from 2 to 12 h at 4 °C prior to incubation with primary antibodies overnight at 4 °C. In addition, the secondary antibodies used in this study were conjugated with Alexa Fluor (488, 568, and 647; Thermo Fisher, 1:200). 4′, 6-diaminodino-2-phenylindole was used as a nuclear counterstain. After staining, tissues were mounted in Fluoro-Keeper antifade reagent, non-hardening type mounting medium (Nacalai, cat. no. 12593-64).

**Proximity Ligation Assay.** The PLA in MBT2 cells and tissues of eyelids was performed using Duolink in situ PLA probe anti-mouse plus (Sigma-Aldrich Cat. No, Duo92001-30RXN), Duolink in situ PLA probe anti-rabbit minus (Sigma-Aldrich Cat. no. Duo92005-30RXN), Duolink in situ detection reagent green (Sigma-Aldrich Cat. no. 92014-30RXN) and Duolink in situ wash buffers, fluorescence (Sigma-Aldrich, Cat. no. Duo82049-4L) according to the manufacturer's instructions. Antibodies used in this assay; mouse anti-GRHL3 (C-12) (Santa Cruz, Cat. no. sc-398838), rabbit anti-USP39 (Sigma-Aldrich, Cat. no. U0385), and rabbit anti-USP39 (Sigma-Aldrich, Cat. no. AV38825).

**Kusabira-Green reporter system.** We used the CoralHue Fluo-chase kit (MBL). N-terminal (KG_N) and C-terminal (KG_C) fragments of the fluorescent protein mKG (monomeric Kusabira Green) are fused to fragments of the mouse *Grhl3* cDNA and

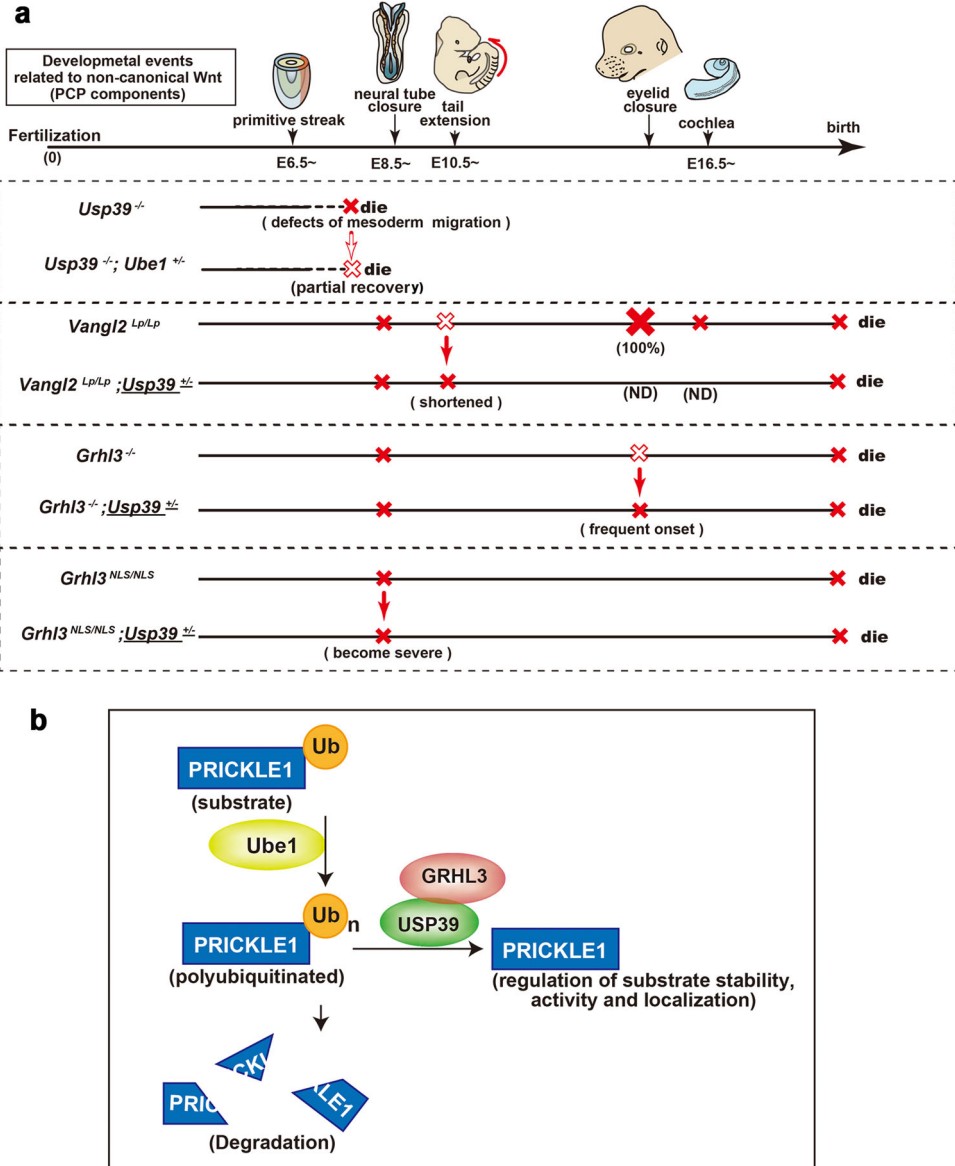

**Fig. 10 Compound mutant phenotypes of *Vangl2*, *Grhl3*, *Usp39*, and *Ube1* genes during mouse development and schematic model of USP39/GRHL3 complex in the de-ubiquitination pathway. a** Schematic illustrations of phenotypes associated with non-canonical Wnt signaling during mouse development: solid red crosses denote defective formation with expression of complete penetrance (100%). Open red crosses denote defective formation but the penetrance is lower or the phenotype is milder. Solid arrows indicate enhancement of the penetrance or severity of defective phenotypes observed by removal of one copy of the *Usp39* gene. The open arrow indicates restoration of the severity of the defect in *Usp39*$^{-/-}$ embryos by removal of one copy of the *Ube1* gene. **b** A schematic model representing how the GRHL3/USP39 interaction can stabilize the PRICKLE1 protein as a target of PCP-related molecules via deubiquitination. ND not determined, PCP planar cell polarity.

mouse *Usp39* cDNA. NIH3T3 cells and MCF7 cells were co-transfected with an equal amount of *mKG_N-mGrhl3* and *mKG_C-mUsp39* plasmids. When the complementary components are co-expressed, the chromophore of mKG is reconstituted and fluoresces.

**In situ hybridization**. Antisense digoxigenin-labeled riboprobes were in vitro transcribed (Roche). For whole-mount in situ hybridization, embryos were fixed in 4% PFA for 2 h at 4 °C, washed three times for 10 min in 1×phosphate buffered saline (1×PBS/0.1% Tween 20 [PBST]), and then dehydrated through a graded series of 25%, 50%, and 75% methanols. Embryos were stored in 100% methanol at −20 °C until in situ hybridization. Whole-mount in situ hybridization was performed as previously described[9].

**Histology**. For histological analysis, embryos and explants were fixed in Bouin's fixative, dehydrated, and embedded in paraplast. Serial sections (8 μm thickness) were generated and stained with hematoxylin and eosin.

**Scanning electron microscopy**. For scanning electron microscopy, tissues were fixed with 4% PFA and 0.25% glutaraldehyde for 24 h at 4 °C in Hank's Balanced Salt Solution (HBSS) buffer (Gibco/Thermo Fisher Scientific; Cat. no. 14065-056), followed by fixation with 1% $O_SO_4$ in HBSS buffer for 1 h at 4 °C. Specimens were then hydrated in a graded ethanol series, transferred into t-butyl alcohol, and dried. Dried specimens were examined under a scanning electron microscope (Hitachi; TM3030 plus).

**Mice**. Animal experiments followed fundamental guidelines for their proper conduct and related activities in academic research institutions under the jurisdiction of the Ministry of Education, Culture, Sports, Science and Technology in Japan, and were approved by institutional committees at the Research Institute for Osaka Women's and Children's Hospital for animal and recombinant DNA experiments (approval numbers: ME−2019-1, ME-2020-1, ME-2021-1, EMB-2019-1, EMB-2020-1, and EMB-2021-1).

For timed breeding, the presence of a vaginal plug indicated embryonic day 0.5 (E0.5) timepoint.

**Generation of *Usp39* and *Ube1* knockout mouse**. The *Usp39* and *Ube1* knockout mouse were generated by the electroporation of preformed CRISPR (cr)RNA/trans-activating crRNA (tracrRNA) (IDT, Coralville, IO, USA; cat. no. 1072533)/Cas9 protein (IDT; cat no. 1074181) complexes. For gRNAs, sequences targeting exon 1 of *Usp39* (IDT; cat. No. 100224849) and exon 6 of *Ube1* (5′- AGAACTCCCCC ACTCGCAGCTGG; IDT) were used. Respective gRNAs were dissolved in Opti-MEM (Thermo Fisher Scientific) and stored at −20 °C until use. Fertilized eggs of CD-1 mice were collected at E0.5 from oviducts of naturally intercrossed females. The fertilized eggs were incubated in KSOM/AA medium until electroporation. CUY21EDIT II (BEX Co. Ltd.) was used in the electroporation. Electroporation conditions were as follows: 30 V, five times of 3 msec of the ON cycle, and 97 msec of the OFF cycle. After electroporation, the zygotes were immediately collected from the M2 medium, followed by washing with KSOM medium two times at 37 °C, and then incubated in 5% $CO_2$ in an incubator until the two-cell stage. Then, two-cell stage embryos were transferred to pseudo-pregnant mice.

**Confocal imaging**. Confocal images were acquired on Fluoview FV1000 and Fluoview FV3000 confocal microscopes (Olympus) using cellSens software (Olympus). Transmitted light images were captured on an Olympus BX53 light microscope (Olympus) with a DP74 digital camera (Olympus).

**Statistics and reproducibility**. Statistical analyses were conducted using Excel (Microsoft) software (Supplementary Data 3). A one or two-tailed, unpaired Student's *t* test was used as described in Supplementary Data 3. $P < 0.05$ was considered significant. The sample numbers analyzed for each experiment are indicated in Table S2.

**Reporting summary**. Further information on research design is available in the Nature Research Reporting Summary linked to this article.

## Data availability

All data generated or analysed during this study are included in this published article, Figs. S1–S15, Table S1, S2, and Supplementary Data 1–3. RNA-seq raw data were deposited with DNA Data bank of Japan (DDBJ) (Accession number: DRA011806, PRJDB11465; Project title: Transcriptomic analysis of wild type versus *Usp39*-deficient mouse embryos submitted by Isao Matsuo and Chiharu Kimura-Yoshida).

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

## Acknowledgements

We are grateful to Drs. Martin, G., Thomas, P., Beddington, R., Schoeler H., and Hogan, B. for plasmids. The monoclonal antibody against keratin, type II/cytokeratin 8 (TROMA-1) was developed by Brulet, P. and Kemler, R., and obtained from the Developmental Studies Hybridoma Bank, created by the National Institute for Child Health and Human Development of the National Institutes of Health and maintained at The University of Iowa, Department of Biology, Iowa City, IA, USA. This work was supported in part by a grant-in-aid for Scientific Research on Priority Areas and on Innovative Areas (JP19H04798), Transformative Research Areas (A) (JP21H05794) and Scientific Research (B) (JP19H03238), (C)(JP19K08291), (C)(JP19K06685), Challenging Research (Exploratory) (21K19467) from the Ministry of Education, Culture, Sports, Science, and Technology, Japan, and by the Naito Foundation, Uehara Memorial Foundation, the NOVARTIS Foundation (Japan) for the Promotion of Science, and Takeda Science Foundation.

## Author contributions

C.K.-Y. and I.M. initiated the research and planned the experiments. C.K.-Y., K.M., S.I.K., and I.M. performed the experiments and analyzed the data. C.K.-Y., S.K., and I.M. contributed to the writing of the manuscript.

## Competing interests

The authors declare no competing interests.
