## [Peer Review File · Communications Biology]

Reviewers' comments:

Reviewer #1 (Remarks to the Author):

Kimura-Yoshida and colleagues investigated the role of the grainyhead-like 3 transcription factor (GRHL3) during epithelial morphogenesis. They find that the nuclear/cytoplasmic shuttling of this protein is associated with canonical and non-canonical PCP Wnt Signaling, respectively. Interestingly, the authors identify the ubiquitin-specific protease 39 (USP39) as a cytosolic partner of GRHL3, and that this can positively contribute to modulating the non-canonical Wnt pathway, hence contributing to epithelial morphogenesis. The authors also show evidence that, in vivo in the mouse, loss of USP39 causes a series of developmental defect which are consistent with a perturbed PCP.

I found the article clearly written, and the findings certainly worth reporting. Below I present a series of comments which, if addressed, would substantiate the findings and, in my opinion, improve the strength and clarity of the article.

1) It is not entirely clear why the authors made the choice of using MCF7 cells, as these do not represent an ideal model for studying epidermal morphogenesis. Perhaps, an explicative sentence in the article, would suffice to clarify this.

2) In Figure 1a, it is not clear how the specific bands identified in the GST-GRHL3 samples, and that are not present in the GST-alone sample, are determined. Why also did the authors decide to focus on USP39 as opposed to other proteins that seem to appear in this group (e.g., HP1BP3)?

3) I find very difficult to understand the VIP assay, and I believe that both a better description and an improved Figure S2 would render more justice to this interesting analysis. This is also true for the figure caption of S2, which in my opinion does not describe well the experiment (and it is also confusing as, for example, the panel d is called before b and c).

4) In commenting Fig S3d, the authors say "Grhl3-full-EGFP was mainly localized in the nucleus irrespective of USP39 knockdown in MCF7 cells". However, few lines later they conclude that "these findings suggest that USP39 is necessary for the correct localization of GRHL3 in the cytoplasm by interacting with the Ub-like domain of GRHL3 in MCF7 cells."

While I see the point that one domain, when isolated, could behave differently from the full-length protein, it still remains true that the latter provides a more important readout, and that it is wrong to conclude that "USP39 is necessary for the correct localization of GRHL3 in the cytoplasm".

5) At page 12, the authors show that simultaneous transfection of b-cateninS37A and Usp39 induced the formation of LM-epidermal cells in the absence of Grhl3 overexpression. Consistently, Usp39 overexpression combined with Wnt agonists yield a similar outcome. But then the authors go on to conclude that "Since both canonical and non-canonical Wnt pathway signaling is necessary for the formation of LM-epidermal cells, and USP39 was able to induce LM-epidermal cells when canonical Wnt signaling alone was activated (Fig. 2h–l), these findings indicate that USP39 can direct epithelial morphogenesis by enhancing the enrichment of actomyosin networks, via activation of non-canonical Wnt signaling."

I disagree with this conclusion, as I believe that the experimental set up successfully allows them to assess the contribution of canonical, but not of the non-canonical Wnt signalling.

6) I consider wrong the sentence that the authors write at page 13, in describing the knockout embryos "and were morphologically distinguishable from wild-type embryos at E6.5". Few lines later, and clearly in figure 3, the authors show that at this stage, Usp39 knockouts fail in forming a primitive streak.

7) The authors identify that PCP markers, VANGL2 and SCRIB are lost in Usp39 knockouts and conclude that "These findings indicate that Usp39 deficiency reduced PCP-related molecules at the protein level". However, Brachyury is an ontogenetically relevant target of the canonical Wnt cascade, and primitive streak formation is regulated by b-catenin and its canonical role. How do the authors reconcile this apparent contradiction – or additional layer of complexity?

8) A general comment applies to all figures, which are sometimes too dense and can benefit for an

improved design aimed at rendering them easier to read and follow.

Reviewer #2 (Remarks to the Author):

Previous work from the authors has already shown that this cytoplasmic localization is essential for epidermal differentiation, when GRHL3 presumably switches from a WNT/CTNNB1 driven role in the nucleus to a WNT/PCP effecting role in the cytoplasm.

This sounds like a worthy phenomenon to follow up, as it is potentially critical for barrier formation in the epidermis and presumably has other roles as well, e.g. neural tube closure.

This paper promises to show that the interaction between GRHL3 and USP39 promotes the cytoplasmic localization of the transcription factor GRHL3. It also promises, in the abstract at least, a hint of a mechanism and a clear demonstration that USP39 affects PCP signaling. After reading the paper, I am not convinced that these claims have been met.

The authors do perform a large set of mouse developmental genetics experiments that show a genetic interaction between USP39 and GRHL3 as well as between USP39 and multiple PCP components in some (e.g. eyelid closure) but not all (e.g. organ of corti) PCP processes. The amount of developmental biology experiments performed is impressive and here all appropriate controls appear to be present.

From a message perspective, however, it is not entirely clear which story the authors are trying to tell and mechanistically resolve. As it stands, they neither demonstrate convincingly just how USP39 causes the cytoplasmic localization of GRHL3 and how that affects its specific function, nor do they clearly demonstrate the role/function of USP39 in PCP. Presumably the two are related, but how is not entirely unclear.

USP39 is hypothesized to both be downstream of WNT signaling and to change the expression/localization of Vangl2 and others AND to act at the level of the actinmyosin network? It is a bit unclear in what model the authors ultimately end up thinking and as a reader I am left behind with many questions after seeing so much data from different systems/tissues, as it is not clear if and how it all fits together in a model that tells us anything new.

Main points:

1. Figure 1: co-localization really cannot be assessed from the images provided at this level of magnification and level of Grhl3 overexpression.
2. Figure 1: Is this the endogenous USP39 signal? Or is USP39 overexpressed? The pattern looks quite different in all images unrelated to GRHL3 expression (e.g. speckled/vesicular in j, membrane ruffles in i, etc.)
3. Can the authors provide a reference for K8 staining being a good marker for epidermal differentiation? Related to this: one really has to be quite familiar with these large epidermal cells in order to make sense of the data in Figure 2. To the less informed reader they are difficult to interpret and many of the channels appear to be overexposed. Moreover, in the control (2j) there is also some k8 expression, so how much of an epithelial differentiation are we really achieving in response to USP39 here? Why don't the authors provide any Krt17/19 stainings as shown in Figure S1 to really identify the LM cells??
4. How do the authors explain the fact that CA-beta-catenin transforms a whole field of cells to Krt8 positive cells, but in the combination with USP39 this is very inefficient and only one LM epidermal cell arises?
5. Figure 3: The loss of VANGL2 and SCRIB is dramatic and quite clear, but can the authors maybe point arrows at the more subtle changes we are supposed to observe in the panel on the right?

Reviewer #3 (Remarks to the Author):

The authors are interested in GRHL3, a transcription factor involved in epithelial cell migration in development. Using a GST-pulldown approach they identify multiple factors including USP39. In a series of biochemical experiments that rely heavily on forced over-expression they suggest USP39 might regulate the localization of GRHL3. Localization studies in addition are hard to interpret due to lack of quantitation. After USP39 knockout in mice, it's not clear if GRHL3 changes localization. Finally, there is a genetic interaction between GRHL3 and both USP39, and the PCP component VANGL2 which looks interesting but is not tightly connected to the initial biochemistry.

Specific comments that may help the authors:

USP39 is commonly isolated in mass spectrometry pull-downs according to the contaminant repository for affinity purification mass spectrometry data (CRAPome V2.0) (<https://reprint-apms.org/?q=reprint-home>). For the author's information, I took the list of identified proteins in Fig 1A and searched in CRAPome. Out of 716 experiments in that database, the identified proteins were found the indicated number of times.

SF3B1 361; XRCC5 428; XRCC6 388; HP1BP3 214; USP39 109

This strongly suggests that USP39, along with the others, is a common contaminant in IP experiments. Given this, it is necessary to have a much more extensive examination of whether the two proteins truly interact at physiological concentrations in cells. The IP of endogenous proteins in Fig 1a is a step in the right direction, but by itself is cursory and lacks rigor. The alternative hypothesis, of course, is that GRHL3 and USP39 only interact at very high and non-physiologic concentrations. Based on the data presented here, I favor this latter hypothesis.

Indeed, forced over-expression is fraught with potential artifacts due to low-affinity interactions, and so it's important to know what the fold-overexpression is in every experiment. This means showing immunoblots using antibodies against the endogenous proteins in every transfection experiment to see if the transfection produces, for example, two-fold, or 100-fold more protein. This level of rigor is lacking here. It also needs to be stated the plasmid quantity per transfection as well. 1 nanogram? 1 microgram?

Fig 1 co-IP needs MW markers.

Four GRHL3 antibodies are used. Which one was used in which experiment, and how were antibodies validated?

Fig leg 1A has a reference to a paper on PALF co-authored by Kanno. I tried to figure out why the figure legend needed footnotes, but it eluded me. And what's the connection of PALF to GRHL3?

Fig 1e the endogenous GRHL3 has multiple bands and the GST-USP39 only binds a subset. Why is that?

Fig 1g – the behaviour of endogenous USP39 looks variable – pan-cellular in some cells, strongly cytoplasmic in others, nuclear-excluded in one. For much of the data in the fluorescence images, it's hard to come to robust conclusions based on images of one or two cells, without quantitation of the images.

Fig S3 and 2d: siRNA is prone to off-target effects and needs multiple controls for validation such as those described in Jackson, A., Linsley, P. Recognizing and avoiding siRNA off-target effects for target identification and therapeutic application. *Nat Rev Drug Discov* 9, 57–67 (2010). doi 10.1038/nrd3010. See specifically the paragraph 'Recognizing and confirming on-target effects.' Such controls must include use of multiple siRNAs for the same target, and/or (the best experiment) rescue of the phenotype by expression of target sequences refractory to siRNA.

The CRISPR-mediated knockout of Grhl3 appears to be in the first exon. Is there any possible downstream methionine that might serve as a re-initiation site?

The authors find a genetic interaction between Grhl3 and Vangl2, and Grhl3 and Usp39. I lack the

expertise to evaluate the images in most of Fig 5. Quantitation and statistics would help support the claims.

REF: COMMSBIO-21-1062-T

USP39 is essential for mammalian epithelial morphogenesis via positively modulating planar cell polarity components (by Kimura-Yoshida et al.)

RESPONSE TO REVIEWER COMMENTS

We thank all three reviewers for their positive and constructive comments on the work presented in our manuscript. We have addressed most of their concerns on a point-by-point basis. Before our detailed reply to each of the reviewers' comments, we would like to first summarize how we have addressed three major issues of concern to the reviewers with regard to the previous version of our manuscript.

Sincerely,

Isao Matsuo, Ph.D.

Chiharu Kimura-Yoshida, Ph.D.

Three major concerns:

(I) Interaction between USP39 and GRHL3 (Reviewers #2 & #3)

Reviewers #2 and #3 asked us to demonstrate whether USP39 can interact with GRHL3 under physiological conditions such as in *in vivo* cells and tissues. To address this issue, we have conducted three new lines of experiments as follows:

First, we have newly exploited a Kusabira-Green reporter system (CoralHue Fluo-chase kit; MBL) that can detect protein-protein interactions as *fluorescent* signals using the protein fragment complementation method (new Fig. 1). When we transfected reporter plasmids carrying protein fragments we found fluorescence only when combinations of two plasmids carrying GRHL3 and USP39 cDNAs were simultaneously expressed in cell lines as shown in the new Figure 1c-1. These additional data explicitly support the notion that USP39 interacts with GRHL3 in living cells.

Additionally, we have analyzed the endogenous subcellular localization of GRHL3 as well as USP39 during mouse eyelid closure (new Fig. 7j-k'') and found that GRHL3 expression appeared to co-locate with USP39 expression in mouse embryos (new Fig. 7j-k'').

To further confirm interaction between USP39 and cytoplasmic-localized GRHL3 in *in vivo* developing embryos, we exploited the *Grhl3^{NLS}* allele (new Fig. 8a; Kimura-Yoshida et al., 2018). In *Grhl3^{NLS}* mutant mice, the nuclear localization sequence is fused with GRHL3 protein and cytoplasmic-localized GRHL3 products are reduced but nuclear-localized GRHL3 products are up-regulated *in vivo* (new Fig. 8b-e). Subsequently, we tested whether *Usp39* could interact with *Grhl3^{NLS}* in terms of neural tube defects, and, consequently, found that neural tube defects of *Grhl3^{NLS}* embryos became more severe by the removal of one copy of the *Usp39* gene (new Fig. 8f-h). This additional finding clearly demonstrates that *Usp39* genetically interacts with cytoplasmic-localized GRHL3 during neural tube closure.

The aforementioned three new lines of evidence are included in the revised manuscript and clearly strengthen our statement that USP39 interacts with GRHL3 *in vivo* during mouse morphogenesis.

(II) USP39 and PCP pathway (Reviewer #1)

Reviewer #1 was concerned about whether USP39 is really involved in the PCP pathway but not canonical Wnt pathway. To verify this issue, we conducted three lines of new experiments as follows:

First, we analyzed cell polarity defects in *Usp39*-deficient embryos that have been reported in *Prickle1*-deficiency (Tao et al., 2009) since PRICKLE1 products were down-regulated in the *Usp39*-deficient embryos (new Fig. 5a,b), and *Prickle1*-deficient embryos failed to correctly form a primitive streak similar to *Usp39*-deficient embryos (Tao et al., 2009; Fig. 4 of this manuscript). Then, we investigated the apico-basal polarity phenotype of *Usp39*^{-/-} embryos (new Fig. 5l–o). *Prickle1*-deficient embryos displayed global distortion of nuclear shape in the epiblast region (Tao et al., 2009), and we found *Usp39*-deficient nuclei had a more spherical shape than those of the wild type (new Fig. 5c–k). Then, we scaled the ratio with a vertical-to-horizontal length for the nucleus in the epiblast of wild-type and *Usp39*^{-/-} embryos (new Figs. 5c–k and S6). We found that wild-type nuclei were ellipsoid and elongated along an apico-basal axis, with an average ratio of vertical-to-horizontal length of 2.42 (new Figs. 5c and S6). In contrast, *Usp39*^{-/-} nuclei were more spherical, with an average vertical-to-horizontal ratio of 1.58 (new Figs. 5c and S6). In addition, similar to *Prickle1*^{-/-} epiblasts (Tao et al., 2009), *Usp39*-deficient embryos displayed a mis-localization of acetylated tubulin as well as protein kinase C (PKC)ζ, crucial factors for apico-basal polarity (new Fig. 5l–o). Moreover, consistent with the evidence that *Prickle1*-deficient embryos have no discernible effect on hair cell polarity in the cochlea (Yang et al., 2017; Liu et al., 2014), apparent abnormalities on hair cell polarity were not evident in *Usp39* mutant cochlea (Fig. S7). Altogether, these findings indicate that *Usp39* mutant embryos show the typical PCP-related phenotype similar to that observed in *Prickle1*-deficient embryos.

Next, to further exclude the possibility that *Usp39* involves canonical Wnt signaling, we newly examined any genetic interaction between *Usp39* and *β-catenin* and found this was not evident in double mutant embryos (new Fig. S9). This finding further supports the idea that USP39 may not contribute to the *β-catenin*-dependent canonical Wnt pathway.

Finally, to verify if USP39 can genetically interact with PCP-dependent GRHL3 function in terms of neural tube defects (new Fig. 8), we exploited the *Grhl3*^{NLS} allele (new Fig. 8a; Kimura-Yoshida et al., 2018). In *Grhl3*^{NLS} mutant mice, a nuclear localization sequence was fused with GRHL3 protein and cytoplasmic-localized GRHL3 products were reduced but nuclear-localized GRHL3 was up-regulated *in vivo* (new Fig. 8b–e) so that PCP-dependent *Grhl3* function was selectively lost. Then, we tested whether *Usp39* could interact with *Grhl3*^{NLS} in terms of neural tube defects and found that those of *Grhl3*^{NLS} embryos became more severe after the removal of one copy of *Usp39* (new Fig. 8f–h). These results clearly demonstrate that *Usp39* genetically interacts with PCP-dependent *Grhl3* function *in vivo*.

These above three lines of new experiments strengthen our hypothesis that USP39 involves the non-canonical Wnt signaling pathway rather than the canonical Wnt-dependent pathway during mouse morphogenesis.

Kimura-Yoshida C, Mochida K, Nakaya MA, Mizutani T, Matsuo I. Cytoplasmic localization of GRHL3 upon epidermal differentiation triggers cell shape change for epithelial morphogenesis. *Nat Commun.* 2018 Oct 3;9(1):4059. doi: 10.1038/s41467-018-06171-8.

Liu C, Lin C, Gao C, May-Simera H, Swaroop A, Li T. Null and hypomorph *Prickle1* alleles in mice phenocopy human Robinow syndrome and disrupt signaling downstream of Wnt5a. *Biol Open.* 2014 Sep 4;3(9):861-70. doi: 10.1242/bio.20148375.

Tao H, Suzuki M, Kiyonari H, Abe T, Sasaoka T, Ueno N. Mouse *prickle1*, the homolog of a PCP gene, is essential for epiblast apical-basal polarity. *Proc Natl Acad Sci U S A.* 2009 Aug 25;106(34):14426-31. doi: 10.1073/pnas.0901332106.

Yang T, Kersigo J, Wu S, Fritsch B, Bassuk AG. *Prickle1* regulates neurite outgrowth of apical spiral ganglion neurons but not hair cell polarity in the murine cochlea. *PLoS One.* 2017 Aug 24;12(8):e0183773. doi: 10.1371/journal.pone.0183773.

(III) Molecular mechanisms of USP39 to control PCP components (Reviewer #2)

To explore molecular mechanisms underlying how USP39 regulates the non-canonical Wnt pathway, we hypothesized that *Usp39* may be involved in two distinct biochemical functions as previously reported: RNA splicing and deubiquitination (Rios et al., 2011; Peng et al., 2020). First, to test whether aberrant RNA splicing is involved in *Usp39* mutant phenotypes, we analyzed mRNA expression profiles between *Usp39*-deficient and wild-type embryos by conducting RNA sequence, mapping, assembly, and alignment (New Supplementary Materials and Methods). Although *Vangl2* expression was apparently reduced at the level of protein but not of mRNA in *Usp39*-deficient embryos (Figs. 4l, l', S5e'), aberrant splicing was not evident in *Vangl2* transcripts from *Usp39*-deficient embryos. Comparative expression profiles revealed that four PCP-related genes, *Dact1*, *Ror2*, *Prickle1*, and *Fzd2*, were also significantly down-regulated in *Usp39*^{-/-} compared to wild-type embryos at E6.5 (fold change > 2, and *p*-value < 0.05; Fig. S13a). However, all four transcripts of these PCP genes appeared to have a normal exon structure without aberrant splicing patterns. Moreover, as *Rb1* transcripts have been shown to display splicing defects by *USP39* mutation in zebrafish (Ríos et al., 2011), such splicing defects were not found in the *Rb1* transcripts of *Usp39*-deficient embryos by alignment of RNA-sequencing data. These results suggest that *Usp39*-deficient phenotypes may not be due to the aberrant splicing of transcripts of PCP components.

Next, we assumed that USP39 may function as a deubiquitinating enzyme for stabilizing PCP components. To explore if USP39 is involved in PCP activity because of its deubiquitinating activity, we newly generated *Ube1* mutant mice by means of CRISPR/CAS9 genome editing to produce *Ube1*;*Usp39* double mutant mice (new Figs. 9, S13). The ubiquitin-protease system is initiated by a ubiquitin-activating enzyme, E1, (UBE1) in mammals (new Fig. S13b). Since *Ube1* is located on the X chromosome, male *Ube1* mutant (*Ube1*⁻) mice displayed early embryonic lethality at an early blastocyst stage (new Figs. 9a–f, S13d). Then, we examined phenotypes of *Usp39*^{-/-};*Ube1*^{+/-} female embryos by crossing *Usp39*^{+/-};*Ube1*^{+/-} female mice with *Usp39*^{+/-};*Ube1*⁺ male mice (new Fig. 9g–r). Marker analysis revealed that in *Usp39*^{-/-} embryos, the primitive streak is not formed at E6.5 but mesendoderm was partially detected at around E7.5 (Fig. 4c–k', new 9h, k, n, q); i.e. mesendoderm cells marked by EOMESODERMIN, *Hex*, and *Foxa2* expression were able to be formed belatedly but unable to migrate toward the anterior side and remained on the posterior side (new Fig. 9k, n, q). In contrast, in *Usp39*^{-/-};*Ube1*^{+/-} embryos, these mesendoderm cells appeared to be able to migrate toward the anterior side compared to those of *Usp39*^{-/-} embryos (new Fig. 9i, l, o, r). These findings demonstrate that failure in *Usp39*^{-/-} mesendoderm migration can be partially restored by the removal of one copy of the *Ube1* gene, i.e. primarily by reducing ubiquitination activity. Therefore, these findings lead us to propose the hypothesis that USP39 contributes to the anterior migration of mesendoderm cells via its deubiquitination activity. To further support this hypothesis, we tested if USP39 activity could control expression of PCP components such as PRICKLE1. Consistent with this hypothesis, PRICKLE1 was actually expressed in mesendoderm cells (new Fig. S13e, f) and, moreover, PRICKLE1 protein was reduced in *USP39* knockdown cells (new Fig. S13g–j). Taken together, these additional findings are included in the revised manuscript and explicitly support the idea that USP39 may participate in the up-regulation of PCP components partly through its deubiquitination activity for mammalian morphogenesis.

Peng Y, Guo J, Sun T, Fu Y, Zheng H, Dong C, Xiong S. USP39 Serves as a Deubiquitinase to Stabilize STAT1 and Sustains Type I IFN-Induced Antiviral Immunity. *J Immunol.* 2020 Dec 1;205(11):3167-3178. doi: 10.4049/jimmunol.1901384

Ríos Y, Melmed S, Lin S, Liu NA. Zebrafish *usp39* mutation leads to *rb1* mRNA splicing defect and pituitary lineage expansion. *PLoS Genet.* 2011 Jan 13;7(1):e1001271. doi: 10.1371/journal.pgen.1001271.

Reviewer #1:

Kimura-Yoshida and colleagues investigated the role of the grainyhead-like 3 transcription factor (GRHL3) during epithelial morphogenesis. They find that the nuclear/cytoplasmic shuttling of this protein is associated with canonical and non-canonical PCP Wnt Signaling, respectively. Interestingly, the authors identify the ubiquitin-specific protease 39 (USP39) as a cytosolic partner of GRHL3, and that this can positively contribute to modulating the non-canonical Wnt pathway, hence contributing to epithelial morphogenesis. The authors also show evidence that, in vivo in the mouse, loss of USP39 causes a series of developmental defect which are consistent with a perturbed PCP.

I found the article clearly written, and the findings certainly worth reporting. Below I present a series of comments which, if addressed, would substantiate the findings and, in my opinion, improve the strength and clarity of the article.

Response: Thank you very much for your constructive comments on our manuscript. We believe that we were able to address most of these concerns as outlined above for the three major concerns (II).

1) It is not entirely clear why the authors made the choice of using MCF7 cells, as these do not represent an ideal model for studying epidermal morphogenesis. Perhaps, an explicative sentence in the article, would suffice to clarify this.

Response: The purpose of our study is to identify a factor controlling the subcellular localization of GRHL3. Therefore, we first screened for cell lines that expressed GRHL3 protein in both the cytoplasm and nucleus using The Human Protein Atlas (<https://www.proteinatlas.org>). We found that only the MCF7 cell line expressed GRHL3 in both the cytoplasm and nucleus. Additionally, since MCF7 is easy to culture, we decided to use MCF7 cells for further study.

As the referee pointed out, MCF7 cells may not be an ideal model for epidermal morphogenesis. Since we have been using ES cells for studies on epidermal morphogenesis, we consider MCF7 as suitable material for the identification of a new binding partner of GRHL3. To confirm our observations, we have newly conducted expression studies in RT4 and MRT2 cells, and incorporated the results into the revised manuscript (Fig. 2).

2) In Figure 1a, it is not clear how the specific bands identified in the GST-GRHL3 samples, and that are not present in the GST-alone sample, are determined. Why also did the authors decide to focus on USP39 as opposed to other proteins that seem to appear in this group (e.g., HP1BP3)?

Response: We identified the specific band for USP39 by means of molecular weight standards. To distinguish proteins that interact with GST-GRHL3 from those of the GST-control, we have shown magnified views of the SDS-PAGE gel in the revised version of the manuscript (new Fig. S2a).

The reason why we narrowed down our search for proteins that interact with GRHL3 to USP39 in this study is as follows: First, with regard to our GST-fusion protein affinity purification experiments conducted by Dr. Kanno, USP39 has rarely been identified in this system according to his extensive experience in this method. Second, GRHL3 has an Ub-like domain that resembles ubiquitin in structure. So, we suspect that USP39 is likely to bind GRHL3 specifically.

3) I find very difficult to understand the VIP assay, and I believe that both a better description and an improved Figure S2 would render more justice to this interesting analysis. This is also true for the figure caption of S2, which in my opinion does not describe well the experiment (and it is also confusing as, for example, the panel d is called before b and c).

Response: We have provided an improved schematic illustration of the VIP assay as requested (new Fig. S3a). This figure more easily outlines the experimental strategy of the VIP assay for the reader's understanding. In addition, we have modified the description regarding the results of the VIP assay in the revised legends of new Figure S3.

4) In commenting Fig S3d, the authors say "Grhl3-full-EGFP was mainly localized in the nucleus irrespective of USP39 knockdown in MCF7 cells". However, few lines later they conclude that "these findings suggest that USP39 is necessary for the correct localization of GRHL3 in the cytoplasm by interacting with the Ub-like domain of GRHL3 in MCF7 cells." While I see the point that one domain, when isolated, could behave differently from the full-length protein, it still remains true that the latter provides a more important readout, and that it is wrong to conclude that "USP39 is necessary for the correct localization of GRHL3 in the cytoplasm".

Response: As suggested, we have corrected our conclusive statement as follows: "Taken together, these findings indicate that USP39 is necessary for the cytoplasmic localization of the Ub-like domain of GRHL3 in RT4 and MCF7 cells. This result suggests that USP39 is involved in the localization of GRHL3 in the cytoplasm (page12, lines 1–4)." This means it might be that if the Ub-like domain is hidden or inactivated by the N-terminal domain of GRHL3 but the Ub-like domain is exposed, USP39 might be able to direct the cytoplasmic localization of full-length of GRHL3.

5) At page 12, the authors show that simultaneous transfection of *b-cateninS37A* and *Usp39* induced the formation of LM-epidermal cells in the absence of *Grhl3* overexpression. Consistently, *Usp39* overexpression combined with Wnt agonists yield a similar outcome. But then the authors go on to conclude that "Since both canonical and non-canonical Wnt pathway signaling is necessary for the formation of LM-epidermal cells, and USP39 was able to induce LM-epidermal cells when canonical Wnt signaling alone was activated (Fig. 2h–l), these findings indicate that USP39 can direct epithelial morphogenesis by enhancing the enrichment of actomyosin networks, via activation of non-canonical Wnt signaling.".

I disagree with this conclusion, as I believe that the experimental set up successfully allows them to assess the contribution of canonical, but not of the non-canonical Wnt signalling.

Response: To avoid confusion, we modified our statement as follows:

We have previously shown that canonical Wnt signaling, such as β -catenin, is insufficient to induce multinucleated, large, and mature (LM-) epidermal cells but non-canonical Wnt signaling, such as *Daam1* cDNA, is essential for induction of LM-epidermal cells (Kimura-Yoshida et al., 2018; Fig.S1a). In this study, we have shown that β -catenin alone is unable to induce LM-epidermal cells but is able to induce smaller immature epidermal cells that are completely different from LM-epidermal cells morphologically and molecularly (Fig. 3b–m; Fig. S1a). Please also compare the cell size of LM-epidermal cells and immature epidermal cells in Figures 3b,g and 3d, respectively; *CAG- β -cateninS37A* cDNA induced solely smaller epidermal cells in Figure 3d (these three panels are shown at an almost identical magnification). In addition, Dicer-substrate siRNA of *Usp39* reduced the number of LM-epidermal cells induced by *Grhl3* cDNA as shown in Figure 3c. Therefore, β -catenin-dependent canonical signaling can induce epidermal differentiation but is insufficient for the induction of LM-epidermal cells for which additional non-canonical Wnt signaling is necessary. Finally, we have newly shown the characteristics of LM-epidermal cells induced by *CAG- β -cateninS37A* cDNA and *Usp39* cDNA with specific molecular markers as shown (new Fig. 3i–m); these cells express mature epidermal markers as well as PCP markers."

These findings together support the hypothesis that USP39 can direct epithelial morphogenesis by activation of non-canonical Wnt signaling. The above data are also summarized in the table below:

Kimura-Yoshida C, Mochida K, Nakaya MA, Mizutani T, Matsuo I. Cytoplasmic localization of GRHL3 upon epidermal differentiation triggers cell shape change for epithelial morphogenesis. *Nat Commun.* 2018 Oct 3;9(1):4059. doi: 10.1038/s41467-018-06171-8.

Expression plasmids	Chemical treatment	caononical Wnt	non-canonical (PCP)	Formation of LM-epi.cells
pgk-neo (negative control)				— (Figure 3n)
Grhl3 cDNA		ON	ON	+ (Figure 3b,S1c)
Grhl3 cDNA	Dicer-substrate siRNA of Usp39	ON	OFF	↓ (reduced) (Figure 3c)
β - cateninS37A cDNA		ON		— (Figure 3d,e)
Usp39 cDNA			ON	— (Figure 3f)
β - cateninS37A cDNA + Usp39 cDNA		ON	ON	+ (Figure 3g-m)
Usp39 cDNA	Canonical Wnt agonist	ON	ON	+ (Figure 3o,p)
Daam1 cDNA			ON	—*
β - cateninS37A cDNA + Daam1 cDNA		ON	ON	+*
β - cateninS37A cDNA + RAC1		ON	ON	+*
	Canonical Wnt agonist + Rac activator	ON	ON	+*

*: Kimura-Yoshida et al., 2018.

6) I consider wrong the sentence that the authors write at page 13, in describing the knockout embryos “and were morphologically distinguishable from wild-type embryos at E6.5”. Few lines later, and clearly in figure 3, the authors show that at this stage, *Usp39* knockouts fail in forming a primitive streak.

Response: To avoid confusion to the reader, we used the term “distinct” instead of “distinguishable” in the revised manuscript (page 15, line 5).

7) The authors identify that PCP markers, *VANGL2* and *SCRIB* are lost in *Usp39* knockouts and conclude that “These findings indicate that *Usp39* deficiency reduced PCP-related molecules at the protein level”. However, *Brachyury* is an ontogenetically relevant target of the canonical Wnt cascade, and primitive streak formation is regulated by β -catenin and its canonical role. How do the authors reconcile this apparent contradiction – or additional layer of complexity?

Response: Please see above for our response regarding one of the major concerns: (II) *USP39* and PCP pathway (for Reviewer #1).

We consider failure in the formation of the primitive streak observed in *Usp39*^{-/-} embryos is due to the migration defects of the mesendoderm and aberrant apico-basal polarity, both of which are mainly controlled by non-canonical Wnt signaling (as shown in new Figs. 5,9,S6). The precise description of *Usp39* mutant phenotypes is described above in our response to major concerns (II).

8) A general comment applies to all figures, which are sometimes too dense and can benefit for an improved design aimed at rendering them easier to read and follow.

Response: Many of the figure panels were corrected, and abbreviations and arrows for non-specialists are newly included in the revised manuscript.

Reviewer #2:

Previous work from the authors has already shown that this cytoplasmic localization is essential for epidermal differentiation, when GRHL3 presumably switches from a WNT/CTNNB1 driven role in the nucleus to a WNT/PCP effecting role in the cytoplasm. This sounds like a worthy phenomenon to follow up, as it is potentially critical for barrier formation in the epidermis and presumably has other roles as well, e.g. neural tube closure.

This paper promises to show that the interaction between GRHL3 and USP39 promotes the cytoplasmic localization of the transcription factor GRHL3. It also promises, in the abstract at least, a hint of a mechanism and a clear demonstration that USP39 affects PCP signaling. After reading the paper, I am not convinced that these claims have been met.

The authors do perform a large set of mouse developmental genetics experiments that show a genetic interaction between USP39 and GRHL3 as well as between USP39 and multiple PCP components in some (e.g. eyelid closure) but not all (e.g. organ of corti) PCP processes.

The amount of developmental biology experiments performed is impressive and here all appropriate controls appear to be present.

From a message perspective, however, it is not entirely clear which story the authors are trying to tell and mechanistically resolve. As it stands, they neither demonstrate convincingly just how USP39 causes the cytoplasmic localization of GRHL3 and how that affects its specific function, nor do they clearly demonstrate the role/function of USP39 in PCP. Presumably the two are related, but how is not entirely clear.

USP39 is hypothesized to both be downstream of WNT signaling and to change the expression/localization of Vangl2 and others AND to act at the level of the actinmyosin network? It is a bit unclear in what model the authors ultimately end up thinking and as a reader I am left behind with many questions after seeing so much data from different systems/tissues, as it is not clear if and how it all fits together in a model that tells us anything new.

Response: Thank you very much for your constructive comments on our manuscript. We believe that we were able to address most of these concerns in the responses to the three major concerns outlined above (I,III).

Main points:

1. Figure 1: co-localization really cannot be assessed from the images provided at this level of magnification and level of Grhl3 overexpression.

Response: We have newly verified our expression studies in a different cell line, RT4 cells, and shown that the Ub-like domain of GRHL3 is located in both the nucleus and cytoplasm but that knockdown of USP39 expression selectively down-regulated cytoplasmic expression of the Ub-like domain of GRHL3 (new Fig. 2n–s). These findings confirm the findings of our previous expression studies in MCF7 cells (now Fig. S4 instead of the prior Fig. 1g–j). In addition, we have also newly used a Kusabira-Green reporter system and analyzed endogenous expression during mouse eyelid closure to show co-localization between full-length GRHL3 and USP39 proteins *in vivo* (new Fig. 1 and 7j–k’). Please see major concerns (I) above.

2. Figure 1: Is this the endogenous USP39 signal? Or is USP39 overexpressed? The pattern looks quite different in all images unrelated to GRHL3 expression (e.g. speckled/vesicular in j, membrane ruffles in i, etc.)

Response: The previous Figure 1f (now Fig. S4a) showed how USP39 was overexpressed in MCF7 cells; however, the previous Figure 1g–j (now Fig. S4b–f) showed endogenous USP39 expression. As this reviewer pointed out, nuclear GRHL3 expression appears to be distinct from endogenous USP39 but the cytoplasmic expression of CP2 or the Ub-like domain of GRHL3 appears to overlap with endogenous USP39 expression in MCF7 cells (now Fig. S4d,e).

3. Can the authors provide a reference for K8 staining being a good marker for epidermal differentiation? Related to this: one really has to be quite familiar with these large epidermal cells in order to make sense of the data in Figure 2. To the less informed reader they are difficult to interpret and many of the channels appear to be overexposed. Moreover, in the control (2j) there is also some k8 expression, so how much of an epithelial differentiation are we really achieving in response to USP39 here? Why don't the authors provide any Krt17/19 stainings as shown in Figure S1 to really identify the LM cells??

Response: In accordance with the reviewer's suggestions, we have newly examined the expression of several crucial markers in LM-epidermal cells induced by *CAG-β-cateninS37A* cDNA and *Usp39* cDNA. Consequently, induced LM-epidermal cells from ES cells expressed mature epidermis markers, such as TFAP2A, TFAP2B, and KRT17/19, and PCP components including SCRIB and phospho-MLC specifically (new Fig. 3i-m). We did not use these specific antibodies for mostly technical reasons: cell morphology as well as glass-bottom dishes tend to be broken during the autoclave step required for antigen activation in immunohistochemistry using these antibodies. It was therefore quite difficult to constantly monitor the expression. In addition to Krt8 (TROMAI) expression, we always focused on the larger cell size and multi-nucleation of LM-epidermal cells morphologically, which is another characteristic of LM-epidermal cells. Please compare the cell size of the two different types of Keratin8-positive epidermal cells (Fig. S1b,c; both panels are an identical magnification). LM-epidermal cells are almost one dozen, at least five-fold larger than simple immature epidermal cells. Finally, to clarify the conclusions drawn from *in vitro* transfection experiments, our results are summarized in the table below:

Expression plasmids	Chemical treatment	caonical Wnt	non-canonical (PCP)	Formation of LM-epi.cells
pgk-neo (negative control)				— (Figure 3n)
Grhl3 cDNA		ON	ON	+ (Figure 3b,S1c)
Grhl3 cDNA	Dicer-substrate siRNA of Usp39	ON	OFF	↓ (reduced) (Figure 3c)
β-cateninS37A cDNA		ON		— (Figure 3d,e)
Usp39 cDNA			ON	— (Figure 3f)
β-cateninS37A cDNA + Usp39 cDNA		ON	ON	+ (Figure 3g-m)
Usp39 cDNA	Canonical Wnt agonist	ON	ON	+ (Figure 3o,p)
Daam1 cDNA			ON	—*
β-cateninS37A cDNA + Daam1 cDNA		ON	ON	+*
β-cateninS37A cDNA + RAC1		ON	ON	+*
	Canonical Wnt agonist + Rac activator	ON	ON	+*

*; Kimura-Yoshida et al., 2018.

4. How do the authors explain the fact that CA-beta-catenin transforms a whole field of cells to Krt8 positive cells, but in the combination with USP39 this is ver inefficient and only one LM epidermal cells arises?

Response: As the reviewer pointed out, canonical Wnt signaling, in particular CA- β -catenin, appears to induce immature Krt8-positive epidermal cells throughout the EB colony; however, additional non-canonical signaling induces a few large and mature (LM-) epidermal cells without immature small epidermal cells. At present, we do not have a clear answer for this phenomenon since in this study we did not focus on the canonical Wnt-dependent formation of small and immature epidermal cells. One possible explanation may be that we have transfected two plasmids simultaneously into ES cells but not in a sequential manner. Therefore, initial PCP activation by USP39 may inhibit β -catenin-dependent induction of immature epidermal cells while only step-wise expression of β -catenin and subsequent USP39 can form LM-epidermal cells.

In addition, two types of epidermal cells are induced at different depths of the EB colony along the Z axis (please see below a three-dimensional [3D]-reconstructed image). We are not able to analyze both types of epidermal cells on the one plane by two-dimensional (2D) imaging for technical reasons due to higher magnification objective lens. Considering that many of the 2D images of LM-epidermal cells tend to lack images of small and immature epidermal cells even if small and immature epidermal cells are present, many of EB colonies, which have central LM-epidermal cells, are considered to have small and immature epidermal cells peripherally, to some extent, as shown below:

LM-epidermal cells are mostly induced in the upper layer of the EB colony centrally while small immature epidermal cells are induced in the lower layer of the EB colony peripherally.

5. Figure 3: The loss of VANGL2 and SCRIB is dramatic and quite clear, but can the authors maybe point arrows at the more subtle changes we are supposed to observe in the panel on the right?

Response: Based on the reviewer's suggestion, we have added arrows and footnotes in the right panels of the previous Figure 3 (now Fig. 4i-k').

Reviewer #3:

The authors are interested in GRHL3, a transcription factor involved in epithelial cell migration in development. Using a GST-pulldown approach they identify multiple factors including USP39. In a series of biochemical experiments that rely heavily on forced over-expression they suggest USP39 might regulate the localization of GRHL3. Localization studies in addition are hard to interpret due to lack of quantitation. After USP39 knockout in mice, it's not clear if GRHL3 changes localization. Finally, there is a genetic interaction between GRHL3 and both USP39, and the PCP component VANGL2 which looks interesting but is not tightly connected to the initial biochemistry.

Response: Thank you very much for your constructive and careful comments on our manuscript. We believe that we were able to address most of these concerns as described above in the three major areas of concern (I).

Specific comments that may help the authors:

USP39 is commonly isolated in mass spectrometry pull-downs according to the contaminant repository for affinity purification mass spectrometry data (CRAPome V2.0) (<https://reprint-apms.org/?q=reprint-home>). For the author's information, I took the list of identified proteins in Fig 1A and searched in CRAPome. Out of 716 experiments in that database, the identified proteins were found the indicated number of times.

SF3B1 361; XRCC5 428; XRCC6 388; HP1BP3 214; USP39 109

This strongly suggests that USP39, along with the others, is a common contaminant in IP experiments. Given this, it is necessary to have a much more extensive examination of whether the two proteins truly interact at physiological concentrations in cells. The IP of endogenous proteins in Fig 1a is a step in the right direction, but by itself is cursory and lacks rigor. The alternative hypothesis, of course, is that GRHL3 and USP39 only interact at very high and non-physiologic concentrations. Based on the data presented here, I favor this latter hypothesis.

Response: First, the contaminant repository from the above CRAPome V2.0 data are derived solely from immuno-affinity purification methods but not the GST-pulldown assay used in this study. Second, we consider that one of main causes of contaminants with GST-pulldown experiments is brought about by the co-precipitation of nucleic acids. To avoid this as much as possible, we always treat cell extracts with benzonase nuclease (Novagen) as described in Methods. Third, considering that de-ubiquitination enzymes that are ubiquitously expressed are very frequently found in these data banks; e.g., USP5; 148, USP7; 201, USP10; 114, USP11; 78, USP14; 144; USP15; 182, USP16; 50, USP22; 45, USP24; 48, USP28; 59, USP36; 77, USP48; 57 and UBA1; 301, as compared to USP39; 109 but clearly less frequent than Keratins such as KRT1; 671, KRT2; 628, and KRT10; 616, which are likely contaminants. A ubiquitination–deubiquitination system is intimately linked to numerous crucial biological processes. In light of this, the USP family should not just be considered contaminants. Rather, the question should be asked as to whether these USP family proteins play important roles in numerous biological processes? Concurrent with this idea, *Usp39* knock-out mutant mice show very clear PCP mutant phenotypes as described above for the three major concerns (II,III; new Figs. 4,5,7,8,9), further supporting the crucial and unique role of USP39 in mammalian development. Fourth, GRHL3 has a Ub-like domain that resembles ubiquitin in its structure. We therefore analyzed the specific interacting domains of the two proteins and demonstrated that the interaction of GRHL3 is indeed dependent on its Ub-like domain (previous Fig. 1d,e; now Fig. S2d,e). So, we believe that USP39 can bind GRHL3 specifically.

Finally, we have conducted two new lines of experiments as outlined for major issues (I) above. We demonstrated that USP39 can interact with GRHL3 under physiological conditions by means of a Kusabira-Green reporter system (CoralHue Fluo-chase kit; MBL), which can detect protein–protein interactions as *fluorescent* signals using a protein fragment complementation method (new Fig. 1). These additional data explicitly support the idea that

USP39 interacts with GRHL3 in two distinct types of living cells. Moreover, we have found that cytoplasmic GRHL3 expression appeared to overlap with the cytoplasmic expression of USP39 *in vivo* during mouse eyelid closure (Figs. 7j–k’).

These several lines of evidence together with showing a genetic interaction between *Grhl3* and *Usp39* explicitly support the hypothesis that GRHL3 binds USP39 in living cells and embryos.

Indeed, forced over-expression is fraught with potential artifacts due to low-affinity interactions, and so it’s important to know what the fold-overexpression is in every experiment. This means showing immunoblots using antibodies against the endogenous proteins in every transfection experiment to see if the transfection produces, for example, two-fold, or 100-fold more protein. This level of rigor is lacking here. It also needs to be stated the plasmid quantity per transfection as well. 1 nanogram? 1 microgram?

We present western blotting data and its quantification below in **Attached data 1**. Lipofection of USP39-RFP products are observed at higher molecular weight as compared with endogenous USP39 protein. Co-transfection of *Grhl3* cDNA and *Usp39-RFP* shows drastic upregulation of *Usp39-RFP* products. In this study, we transfected a total of 2.5 µg of plasmid sample for each experiment in a 6-well culture dish. This description is included in the revised Materials and Methods (page 32, lines 25–27).

Specific points:

1) Fig 1 co-IP needs MW markers.

Four GRHL3 antibodies are used. Which one was used in which experiment, and how were antibodies validated?

Response: We have added the MW markers as requested for the previous Figure 1 (now Fig. S2a).

We confirmed that three GRHL3 antibodies were used in this study. This included two polyclonal antibodies against aa195–211 and aa478–493 of GRHL3, which are used for immunohistochemistry and have been verified with two independent *Grhl3*-siRNAs in MCF7 cells (Fig. S1e,f,h,i). Additionally, we had previously shown the verification of these antibodies in *Grhl3*-deficient embryos (Kimura-Yoshida et al., 2015; 2018). With respect to the monoclonal antibody against aa435–459 of GRHL3 (Santa Cruz Biotechnology, C-12, sc-398838), which is used for western blotting, this has also been verified previously (Di Girolamo et al., 2016).

Di Girolamo D, Ambrosio R, De Stefano MA, Mancino G, Porcelli T, Luongo C, Di Cicco E, Scalia G, Vecchio LD, Colao A, Dlugosz AA, Missero C, Salvatore D, Dentice M. Reciprocal interplay between thyroid hormone and microRNA-21 regulates hedgehog pathway-driven skin tumorigenesis. *J Clin Invest*. 2016 Jun 1;126(6):2308-20. doi: 10.1172/JCI84465.

Kimura-Yoshida C, Mochida K, Ellwanger K, Niehrs C, Matsuo I. Fate Specification of Neural Plate Border by Canonical Wnt Signaling and *Grhl3* is Crucial for Neural Tube Closure. *EBioMedicine*. 2015 Apr 18;2(6):513-27. doi: 10.1016/j.ebiom.2015.04.012.

Kimura-Yoshida C, Mochida K, Nakaya MA, Mizutani T, Matsuo I. Cytoplasmic localization of GRHL3 upon epidermal differentiation triggers cell shape change for epithelial morphogenesis. *Nat Commun*. 2018 Oct 3;9(1):4059. doi: 10.1038/s41467-018-06171-8.

2) Fig leg 1A has a reference to a paper on PALF co-authored by Kanno. I tried to figure out why the figure legend needed footnotes, but it eluded me. And what’s the connection of PALF to GRHL3?

Response: We have removed this reference from the reference list.

3) Fig 1e the endogenous GRHL3 has multiple bands and the GST-USP39 only binds a subset. Why is that?

Response: In MCF7 cells, mRNA expression appears to be high compared to its corresponding protein expression (The Human Protein Atlas; <https://www.proteinatlas.org/>). Thus, we consider that GRHL3 itself may be rapidly degraded in *Grhl3*-transfected MCF7 cells and USP39 may bind some of the degradation products of GRHL3.

4) Fig 1g – the behaviour of endogenous USP39 looks variable – pan-cellular in some cells, strongly cytoplasmic in others, nuclear-excluded in one. For much of the data in the fluorescence images, it's hard to come to robust conclusions based on images of one or two cells, without quantitation of the images.

Response: Yes. As pointed out by this reviewer, USP39 expression itself appears to be variable. However, this is the endogenous USP39 expression observed in MCF7 cells. Our most important finding is that the behavior of the Ub-like domain in GRHL3 appears to colocalize with USP39 expression in cells. Additionally, other additional images of overlapping expression between the Ub-like domain of GRHL3 and USP39 in MCF7 cells are shown as **Attached data 2**.

5) Fig S3 and 2d: siRNA is prone to off-target effects and needs multiple controls for validation such as those described in Jackson, A., Linsley, P. Recognizing and avoiding siRNA off-target effects for target identification and therapeutic application. *Nat Rev Drug Discov* 9, 57–67 (2010). doi 10.1038/nrd3010. See specifically the paragraph 'Recognizing and confirming on-target effects.' Such controls must include use of multiple siRNAs for the same target, and/or (the best experiment) rescue of the phenotype by expression of target sequences refractory to siRNA.

Response: To address this issue, we have newly conducted USP39 knockdown experiments with Dicer-substrate siRNA (27mer), which is longer than conventional siRNAs (21mer) and has been demonstrated to have higher knockdown activity with strict specificity (Kim et al., 2005; new Fig. 2j–m, 2p–s, 3c, S13i, j). In addition, we also newly present data on multiple independent siRNAs with regard to GRHL3 expression studies (new Fig. S1e, f, h, i).

Kim DH, Behlke MA, Rose SD, Chang MS, Choi S, Rossi JJ. Synthetic dsRNA Dicer substrates enhance RNAi potency and efficacy. *Nat Biotechnol.* 2005 Feb;23(2):222-6. doi: 10.1038/nbt1051.

6) The CRISPR-mediated knockout of *Grhl3* appears to be in the first exon. Is there any possible downstream methionine that might serve as a re-initiation site?

Response: We have analyzed the mutant mRNA sequence derived from *Usp39*^{-/-} embryos (the F0-8 line; 2-bp deletion allele that is mainly used in this study) with RNA-seq mapping and alignment. We found that the entire mRNA sequence of mutant *Usp39* is exactly identical with that of the wild type, except for the 2-bp deletion, while the wild-type sequence is 2209 nucleotides long. Since we have shown the translated amino acid sequence of the mutant allele as **Attached data 3** below, the frame of this 2-bp deletion mutant shifted and has a stop codon instead of the 17th amino acid from the translational start codon (**Attached data 3**). Since the next methionine is only found after the 3'-UTR of *Usp39* (endogenous stop codon appeared at around 1670 nt), any downstream methionine will not be used as an endogenous frame of the USP39 protein.

7) The authors find a genetic interaction between *Grhl3* and *Vangl2*, and *Grhl3* and *Usp39*. I lack the expertise to evaluate the images in most of Fig 5. Quantitation and statistics would help support the claims.

Response: We have demonstrated the genetic interaction between *Usp39* and *Grhl3* **quantitatively** and **statistically** as shown in the previous Figure 5d and now Figure 7d; the penetrance of eyelid defects in *Grhl3*^{-/-}; *Usp39*^{+/-} embryos is much higher than those of *Grhl3*^{-/-} embryos. The expression patterns of Fig. 7l-n' reveal that three PCP-related markers are **undetectable** in *Grhl3*^{-/-}; *Usp39*^{+/-} embryos as compared to the wild type during epithelial morphogenesis. Therefore, it is quite difficult to show quantitative data for no expression (Fig. 7l-n'). For the reader's better understanding, including those without a background in anatomy, we have added arrows and footnotes in the previous Figure 5 (now Fig. 7) of the revised manuscript.

Attached data 1

CAG-Grh13 cDNA and *CAG-Usp39-RFP* cDNA were transfected into mouse ES cells in a 6-well culture dish with Lipofectamine LTX. Thereafter, ES cells were differentiated via embryoid bodies as described in M&M.

(a) Immunoblots with antibodies against USP39 and GAPDH (for normalization control).

(b) Quantification of signal intensity of immunoblots after normalization with GAPDH signals.

Three independent experiments were performed.

Attached data 2

CAG-Grhl3 Ub like- EGFP

Cytoplasmic localization of EGFP-fused Ub-like domain of GRHL3 tends to overlap with the endogenous localization of USP39 (white arrowheads).

[GENETYX-MAC: Translation of Nucleotides into Amino Acids for Thesis]

Date : 2021.11.10
Filename : Untitled
Sequence size : 2207
Sequence Position: 1 - 2207

Translation Position: 1 - 2207

Genetic Code: Standard Genetic Code

GA' 2nt deletion

10 20 30 40 50 60
ATGTCTAGCCGGTCCCAAGCGGACGTCCATGGTTCACCCGGCGAGCGTGAATCCCGAGTC
M S S R S K R Q S H G S T R G A * V R V

Stop

70 80 90 100 110 120
TCGTGGCAGCTCGGTCCGATCAAGAAGGAGCGAGCCGTGAGAAGGAGCCGAGGCGGC
S W Q L G S H Q E G A R P * E G A R G G

130 140 150 160 170 180
GAGCTCCGGGGTGAAGCCGCTCCGCTGAAGCGGGAGGCCGAGCCGGCTCGCGGAGAGGT
E L P G * P G P P R E A G G R A G C A R G

190 200 210 220 230 240
CCGGGCGCCCGCTCCCGGTGTGCGGGTGAAGCGTGAAGCGGAGCCGATGAAGACT
P G A R A P G R A G E A * A R G R * R L

250 260 270 280 290 300
GGAGCCGAGCGGGAGGTGCGAGCAAGAATGCCGAGTGGATTCTGAAGACCGGAGGAG
G A R A G G A S K E W P S G F * R P E E

310 320 330 340 350 360
TCGTCACTGCCGACTTGGATACCATTAATAGGAGTGTCTGCACTTCGACTTTGAGAA
S S L P V L G Y H * * E C A G L R L * E

370 380 390 400 410 420
ACTCTGCTCATTCTCTCCACATCAAGCGATACGCTGTCTGGTGTGGCAAGTA
T L L H F S L A H Q R I R L S G V W Q V

430 440 450 460 470 480
CTTTCAGGCGGGCTTAAAGTCTCATGGCTACATCCAGTGTCCAGTTCAAGCACCA
L S R P G L K V S C L H P Q C P V Q P P

490 500 510 520 530 540
TGTCTTCCTCAACTCCGACACTCTCAAGTTTACTGCTTCTGACAACTATGAATCAT
C L P Q P P H S Q V L L P S * Q L * N H

550 560 570 580 590 600
TGATTCTCGCTGGAGGATCACGTATGTGTGGAAGCTACTTTCAAAAGCAAAAT
* F L A G G Y H V C V E A Y F H K A A N

610 620 630 640 650 660
TGCAAACTGGATAAGCCAAATGTCGGGGTCAATGATGGCACCCTTACCTCGCC
C K L G * A S Q I V P G I * W H H L P A

670 680 690 700 710 720
AGGGATCGTGGGACTGAACAACATAAAGCCAACTGCAAAATGCTGCTGCTGCGAGC
R D R G T E Q H K S Q R L C K C C A A G

730 740 750 760 770 780
TCTATCTAATGTCCTCTTCGGAACACTTCTGGAGGAAGACAATTAACAAGAAT
S I * C P S S S E L L P G G R Q L Q E H

790 800 810 820 830 840
CAAGGCTCTCCGGGGACATCATGTTCTGTGGTTCAGCGTTTGGAGAGCTGATGAG
Q A S S G G H H V L V G S A F W R A D E

850 860 870 880 890 900
AAAGCTCTGGAAACCCCAAACTCAAGGCGCATGCTCTCTCATGAGATGCTTCAGCC
K A L E P P K L Q G A C L S S * D A S G

910 920 930 940 950 960
TGTGTAATTTGACGCAAGAACCTTTCCAAATTAACAACAAGGGGATGGAGTGAATT
C R T L Q Q E D F P N Y Q T R G W S * L

970 980 990 1000 1010 1020
CCTGCTGGTTTCTGAATGCTGCACTGCTCTGGGAGGCCAAGAAGAAGAAAA
P V L V S E C S A L C S G R H Q E E E K

1030 1040 1050 1060 1070 1080
GACTATTGTAATGATGTTTTCCAGGGATCAATGAGAATTTACCAAAAAGCTTCCTCA
D Y C E * C F P G I N E N F H Q K A S S

1090 1100 1110 1120 1130 1140
TCCTGATCTGCCAGCGGAAGAGAAAGCAGCTCTCCACAATGATGAGTACCAAGAGAC
S * S A S S G R E R A A A P Q * * V P R D

1150 1160 1170 1180 1190 1200
GATGGTAGAGTCCAGTTCATGTACTGACCTGGACCTCCACGGCCCGCTGATATA
D G R V H V H V P D A G P S H G P A L *

1210 1220 1230 1240 1250 1260
GGATGAGAAGGAGCTCATCATCCCCAGGTGCCCTCTTCAACATCTGGCCCAAGTT
G * E G A A H H P P G A S L Q H P G Q V

1270 1280 1290 1300 1310 1320
CAACGGCATCCGGGAGAAAGTAACAAGACTTATAAGGAGAACTTCGTAAGCCCTTCCA
Q R H H G E G I Q D L * G E L P E T L P

1330 1340 1350 1360 1370 1380
GCTCACCAAGTCCCTCCGATCTAATCTTTTGCATCAAGGATTACTAAGAACAACTT
A H Q A A S V S N L L H Q E I Y * E Q L

1390 1400 1410 1420 1430 1440
CTTTGGGAGAAATCCAACAATGTCAACTCCCTACCAATGGAATCTGAGAGAGA
L C G E E S N N C Q L P Y H K C G S E R

1450 1460 1470 1480 1490 1500
ATACTTATCTGAAGAAGTCCAAGCCCTCCACAAGAACCCACCTATGATCTCATGCCAA
I L I * R S P S R P Q E H H L * S H R Q

1510 1520 1530 1540 1550 1560
CATCTGCTGATGGCAAGCCCTGAGGGCTCTACAGGATCCACGTCTTCATCATGG
H R A * W Q A L * G L L Q D P R A S S W

1570 1580 1590 1600 1610 1620
GACTGGCAAGTGGTATGAATACAAGACTCCAGGTGACAGACATCTCGCCCAAGTAD
D W Q V V * I T R P P G D R H S A P D

1630 1640 1650 1660 1670 1680
CACACTGTCTGAGGCTACATTGAGATTTGGAAGGGCGGCAATGAGAAACCA
H T V * G V H S D L E E A G Q * * N Q P

1690 1700 1710 1720 1730 1740
GCAGGGGCTTGAAGCCTTGGTGGTTTTTATCCCAAGGGCTGGCTGAAAGTGTA
A G G L N A L L G F Y S Q G L W L K M V

1750 1760 1770 1780 1790 1800
AATAAAATACTGAAGCTGTCTGCAACAGACTGACTGAAAGATTCTGCGCCAGTCC
N K N T E A V A E H R L T E R F L P Q S

1810 1820 1830 1840 1850 1860
AGTTTTATGATTCTGGCTGATCACCATGAAGGAGCTGTCTGCCCTGATGTTCTCGG
S F Y G F W L H H H E G A C L P * C S A

Stop

1870 1880 1890 1900 1910 1920
CTGCTCCGGCAGCAGTGTGGTGGCACTCTCCCTTTCAGCATCAGCAGGCGAAGCT
L S P A A R C G R H S S L S A S A G R T

1930 1940 1950 1960 1970 1980
CTTACAGATGGATGCTTATTTGAATGCTCAAGTATCAAAGAACAGACAGTTCGGTAG
L T D G C L I * I A A S I K E Q T V R *

1990 2000 2010 2020 2030 2040
AAACGTCAAGTCTGGAGCCCTAGGAGTAAACGTGAAGTCAAGCAGCAGCTCTGTCATA
K R Q V W R P * E * T * S Q Q Q P S V I

2050 2060 2070 2080 2090 2100
GGTAGTGGCTTCTGTAGCTCCCTTCACTCTCTGGGAGCGTGGCATGGATTAATA
G S G F P V A P F I S C G T C A N D * I

Stop

2110 2120 2130 2140 2150 2160
TCCTCTGTTTTAAACAGGCTGTGTTTCATATACCCATTGAGGCTTGTAAAGGTT
S S C F * T R L L F H I P H * G L L R V

2170 2180 2190 2200
TTGTTTCTCTTAACTTATATATACTCTGCTGATGCTG
L F S F L T L Y I L L S V V ??

Endogenous STOP codon of mouse Usp39

RNA sequences of mutant *Usp39* allele (F0-8 line /c46_47 del) derived from *Usp39*^{-/-} embryos.

Reviewers' comments:

Reviewer #1 (Remarks to the Author):

The authors have satisfactorily addressed all my concerns.

Reviewer #2 (Remarks to the Author):

The data added by the authors have certainly increased the strength and quality of the manuscript.

As before, the developmental genetics experiments are of high quality – My prior questions regarding the model that the authors proposed hasn't been answered entirely and I still miss a figure summarizing the molecular/biochemical mechanism the authors think is involved.

Perhaps the abstract just appears to promise more of a molecular/biochemical mechanism than the in vivo data are able to provide. I don't think that this is necessarily a problem, it just has to do with the expectancy that the authors themselves create.

Regarding the major concerns that the authors claim to have addressed:

Interaction between USP39 and GRHL3

- The concern was whether these proteins interacted at endogenous levels as much as whether this is true in living cells. This is not solved by the Kusabiro-Green reporter system as this still relies on overexpression.

- "appears to co-localize" also doesn't support a direct interaction as co-localization doesn't mean 'direct interaction' or 'present in the same complex' – also not for the new images that were added. This really requires higher resolution/larger magnification and/or co-localization analysis of imaging data (and I know this is tricky).

- A genetic interaction with the Grhl3NLS allele also doesn't say anything about a biochemical interaction.

The authors did offer more of a molecular mechanism by dissecting that it is likely the deubiquitination activity of USP39 that could account for the upregulation of PCP components – although why they would opt to solely demonstrate this in vivo using mouse models and not via direct biochemical evidence is not entirely clear to me.

Reviewer #3 (Remarks to the Author):

This revision contains an enormous amount of work with crosses of multiple mouse strains and detailed genetic analysis. The biochemical analysis remains problematic.

In the initial review, I noted "It is necessary to have a much more extensive examination of whether the two proteins truly interact at physiological concentrations in cells."

In the revision, the authors have done a series of experiments with over-expressed proteins or fragments, but do not show interaction at physiological concentrations in experiments with robust controls. Co-localization data is not a reliable demonstration of interaction. The lack of direct data on endogenous protein interaction in the revision I must therefore interpret as "they tried and were unable to see such interactions." This does not negate the genetic interactions, but I don't have high confidence that there is physical interaction.

The authors should indicate in every experiment (both text and figure legends) which proteins are endogenously expressed, and which proteins and constructs have forced over-expression. This is currently not clear and makes it hard to understand the experiments.

REF: COMMSBIO-21-1062A (R2)

RESPONSE TO REVIEWER COMMENTS

We thank the three reviewers for their positive and constructive comments on the work presented in our revised manuscript. We have addressed their concerns on a point-by-point basis as follows:

Sincerely,

Isao Matsuo, Ph.D.

Chiharu Kimura-Yoshida, Ph.D.

Reviewers' comments:

Reviewer #1:

The authors have satisfactorily addressed all my concerns.

Response:

We thank this reviewer very much for supporting our manuscript.

Reviewer #2:

The data added by the authors have certainly increased the strength and quality of the ript.

As before, the developmental genetics experiments are of high quality – My prior questions regarding the model that the authors proposed hasn't been answered entirely and I still miss a figure summarizing the molecular/biochemical mechanism the authors think is involved.

Perhaps the abstract just appears to promise more of a molecular/biochemical mechanism than the in vivo data are able to provide. I don't think that this is necessarily a problem, it just has to do with the expectancy that the authors themselves create.

Regarding the major concerns that the authors claim to have addressed:

Interaction between USP39 and GRHL3

- The concern was whether these proteins interacted at endogenous levels as much as whether this is true in living cells.

This is not solved by the Kusabiro-Green reporter system as this still relies on overexpression.

- "appears to co-locate" also doesn't support a direct interaction as co-localization doesn't mean 'direct interaction' or

'present in the same complex' – also not for the new images that were added. This really requires higher resolution/larger magnification and/or co-localization analysis of imaging data (and I know this is tricky).

- A genetic interaction with the Grhl3NLS allele also doesn't say anything about a biochemical interaction.

The authors did offer more of a molecular mechanism by dissecting that it is likely the deubiquitination activity of USP39 that could account for the upregulation of PCP components – although why they would opt to solely demonstrate this in vivo using mouse models and not via direct biochemical evidence is not entirely clear to me.

Reviewer 3:

This revision contains an enormous amount of work with crosses of multiple mouse strains and detailed genetic analysis. The biochemical analysis remains problematic.

In the initial review, I noted "it is necessary to have a much more extensive examination of whether the two proteins truly interact at physiological concentrations in cells."

In the revision, the authors have done a series of experiments with over-expressed proteins or fragments, but do not show interaction at physiological concentrations in experiments with robust controls. Co-localization data is not a reliable demonstration of interaction. The lack of direct data on endogenous protein interaction in the revision I must therefore interpret as "they tried and were unable to see such interactions." This does not negate the genetic interactions, but I don't have high confidence that there is physical interaction.

The authors should indicate in every experiment (both text and figure legends) which proteins are endogenously expressed, and which proteins and constructs have forced over-expression. This is currently not clear and makes it hard to understand the experiments.

Response:

The above two reviewers requested us to prove a direct interaction between USP39 and GRHL3 proteins at physiological concentrations. First, we would like to emphasize that as previously shown in the right bottom panel of Figure S2a, a co-immunoprecipitation experiment with a GRHL3 antibody was conducted in MCF7 cells without any expression of a tag or forced transgenes (Figure S2a). This result indicates that endogenous USP39 co-immunoprecipitated with endogenous GRHL3 in a relatively natural state; supporting protein–protein interaction between GRHL3 and USP39 would be a physiologically relevant interaction.

Second, we have newly analyzed co-localization of USP39 and GRHL3 in MBT2 and periderm cells during mouse eyelid closure by immunohistochemistry using specific antibodies (new Figure 7h–n; 8a–d). Notably, by means of “Amira software”, high-resolution three dimensional-reconstructed images clearly showed that GRHL3 expression mainly overlapped USP39 expression in the cytoplasm (new Figure 7n, 8d).

Third, in order to further strengthen our hypothesis that both proteins can interact at physiological concentrations in individual cells, we have exploited Duolink Proximity Ligation Assay (PLA), which allows us to detect protein–protein interactions *in situ* (at distances < 40 nm) at endogenous protein levels with high sensitivity and specificity (new Figure 8e; Fredriksson et al., 2002; Söderberg et al., 2006). Consequently, we have found that Duolink-amplified signals were evident in the cytoplasm of MBT2 cells (new Figure 8g [magenta]). Moreover, Duolink signals were observed in the cytoplasm adjacent to the nucleus in mouse periderm cells (new Figure 8j,k [yellow]). Given that protein length can generally be considered to be about 10 nm, both proteins were in very close proximity at physiological concentrations in both MBT2 and periderm cells during mouse eyelid closure. These findings further support our genetic interaction data concerning USP39 and GRHL3 during mouse embryogenesis.

As pointed out by the reviewers, however, we also agree with their claim that even such additional data might not completely demonstrate that this interaction is direct and that both proteins exist in the same protein complex. Therefore, we have removed the text describing “GRHL3 and USP39 are in the same protein complex and their physical interaction is direct” and outlined the limitations of our study in the discussion of the revised manuscript as follows: “Specifically, USP39 and GRHL3, at physiological concentrations, co-localized in close proximity (within 40 nm) in MCF7 and periderm cells during eyelid closure (Fig. 8). However, further biochemical studies will be necessary to elucidate whether the physical interaction between the two proteins is direct and that they are present in the same complex in these cells.” (Page 21, lines 4–8)

In order to avoid confusion, we have also included a statement on biochemical and cell culture experiments in which proteins were endogenously expressed or forcibly overexpressed in the figure legends together with Table S4, representing transfected transgene constructs for each experiment, in the revised manuscript as requested by Reviewer #3; i.e., no transfectants indicate the endogenous expression.

Finally, as suggested by Reviewer #2, we have also included a new schematic model showing how PCP activity is involved with USP39 and GRHL3 via deubiquitination. The additional characterization of *Ube1*-deficient embryos is also shown, supporting how UBE1 is involved in ubiquitination in early mouse embryos (new Figures 11b and S13h–m).

Fredriksson S, Gullberg M, Jarvius J, Olsson C, Pietras K, Gústafsdóttir SM, Ostman A, Landegren U. Protein detection using proximity-dependent DNA ligation assays. *Nat Biotechnol.* 2002 May;20(5):473-7. doi: 10.1038/nbt0502-473. PMID: 11981560

Söderberg O, Gullberg M, Jarvius M, Ridderstråle K, Leuchowius KJ, Jarvius J, Wester K, Hydbring P, Bahram F, Larsson LG, Landegren U. Direct observation of individual endogenous protein complexes *in situ* by proximity ligation. *Nat Methods.* 2006 Dec;3(12):995-1000. doi: 10.1038/nmeth947. Epub 2006 Oct 29. PMID: 17072308

REVIEWERS' COMMENTS:

Reviewer #2 (Remarks to the Author):

The authors have addressed all of my previous comments.

Reviewer #3 (Remarks to the Author):

Two reviewers made it clear that the genetic data was sound and the physical interaction data was problematic. The authors have made a limited attempt to modify the text and added a proximity ligation assay, which again does not make a strong case the interactions are direct.

I can only recommend this paper for publication if all of the following text revisions are made:

Abstract: remove the words "as the interacting protein"

Statement of significance: replace "as a GRHL3-interacting protein" with "as a potential GRHL3-interacting protein"

Introduction p 5 Change "we identified USP39 protein as an interacting mediator of GRHL3" to "we identified USP39 protein as a mediator of GRHL3"

Top of page 6 "we identified protein complexes specifically interacting with GRHL3" change to "we identified protein complexes interacting with GRHL3"

Modify "USP39 was subsequently determined to be a GRHL3-interacting protein" to read "USP39 was subsequently determined to be a potential GRHL3-interacting protein, although according to the Contaminant Repository for Affinity Purification, USP39 is a 'sticky' protein found in ~15% of the protein pulldowns in that repository" (and reference the repository)

Change "These findings further support the idea that USP39 is able to bind GRHL3 in living cells" to "These findings further support the idea that endogenous USP39 might bind GRHL3 in living cells"

Change "These biochemical studies demonstrate that GRHL3 can interact with USP39 via the CP2/Ub-like domains of GRHL3, and the ZnF domain of USP39." To "These biochemical studies demonstrate that over-expressed GRHL3 can interact with USP39 via the CP2/Ub-like domains of GRHL3, and the ZnF domain of USP39."

Change "These aforementioned findings, together with biochemical results, suggest that USP39 interacts with GRHL3 in the cytoplasm mainly through the Ub-like domain of GRHL3." To "These aforementioned findings, together with biochemical results, suggest that over-expressed USP39 interacts with GRHL3 in the cytoplasm mainly through the Ub-like domain of GRHL3."

Discussion: change "Current studies indicate that USP39 can interact with the Ub-like domain of GRHL3 for proper localization of GRHL3 in the cytoplasm" to "Current studies, albeit highly reliant on over-expression, indicate that USP39 can interact with the Ub-like domain of GRHL3 for proper localization of GRHL3 in the cytoplasm"